# Smart thrombosis inhibitors without bleeding side effects via charge tunable ligand design

Chanel C. La [1,2], Stephanie A. Smith [3], Sreeparna Vappala [1,4], Reheman Adili[5,13], Catherine E. Luke [6], Srinivas Abbina [1,4], Haiming D. Luo [1,2], Irina Chafeeva[1,4], Matthew Drayton[1,4], Louise A. Creagh[7,8], Maria de Guadalupe Jaraquemada-Peláez [2], Nicole Rhoads [9], Manu Thomas Kalathottukaren[1,4], Peter K. Henke [6], Suzana K. Straus [2], Caigan Du[10], Edward M. Conway [1,4,11,12], Michael Holinstat [5], Charles A. Haynes[7,8], James H. Morrissey [3] ✉ & Jayachandran N. Kizhakkedathu [1,2,4,12] ✉

Current treatments to prevent thrombosis, namely anticoagulants and platelets antagonists, remain complicated by the persistent risk of bleeding. Improved therapeutic strategies that diminish this risk would have a huge clinical impact. Antithrombotic agents that neutralize and inhibit polyphosphate (polyP) can be a powerful approach towards such a goal. Here, we report a design concept towards polyP inhibition, termed macromolecular polyanion inhibitors (MPI), with high binding affinity and specificity. Lead antithrombotic candidates are identified through a library screening of molecules which possess low charge density at physiological pH but which increase their charge upon binding to polyP, providing a smart way to enhance their activity and selectivity. The lead MPI candidates demonstrates antithrombotic activity in mouse models of thrombosis, does not give rise to bleeding, and is well tolerated in mice even at very high doses. The developed inhibitor is anticipated to open avenues in thrombosis prevention without bleeding risk, a challenge not addressed by current therapies.

Venous and arterial thromboembolic disorders such as pulmonary embolism, myocardial infarction and stroke are leading causes of morbidity and mortality in the Western world[1]. In spite of significant advances in antithrombotic and anticoagulant therapies, the problem still persists and improved therapeutic approaches are required. Current therapies are complicated by bleeding side effects as they interrupt key proteases or pathways involved in hemostasis. It is important to maintain a balance between thrombosis and hemostasis, particularly in the design of antithrombotic therapeutics[2,3].

Recently, biological polyanions that trigger the contact pathway of coagulation have been identified[4–6]. As the contact pathway is not required for hemostasis, its involvement in thromboembolic diseases provides promising opportunities for antithrombotic drug design[7–9]. The contact pathway is involved in the activation of inflammation and provides an important link between coagulation and immune-mediated reactions. This has been exemplified in the case of polyphosphate (polyP), a linear polymer of inorganic anionic phosphate[10]. PolyP is a potent procoagulant and proinflammatory molecule[11–13] acting at multiple steps in the coagulation cascade: it triggers clotting via the contact pathway[11,12,14], accelerates factor V activation[11], enhances fibrin clot structure[15,16], and accelerates factor XI back-activation by thrombin[17], depending on the chain length of polyP[14]. Long chain

polyP, released by bacteria during infection, is a potent activator of the contact pathway of coagulation, able to support auto-activation of FXIIa and promote thrombin generation. Platelet-released short chain polyP is reported to assemble as nanoparticles on the cell surface, potentially able to activate the contact pathway[18]. Moreover, polyP is incorporated into fibrin clots, inhibiting fibrinolysis and resulting in clots with reduced stiffness and deformability[15,19]. It is also known that patients with defects in the production of polyP are protected against thrombosis[12].

As a potent accelerator of blood coagulation which is not involved in essential pathways of coagulation, polyP is a promising target for developing therapies to prevent thrombosis with minimal bleeding side effects[11,15,20–22]. Previously, the prevention of polyP's procoagulant activity had been explored through an enzyme-mediated cleavage approach using alkaline phosphatase, a homodimeric enzyme having the physiological role of dephosphorylating compounds[23]. Particularly, *Escherichia coli* exopolyphosphatase (PPX), a cytoplasmic polyphosphatase, catalyses the hydrolysis of intracellular polyP[18,24–26]. While these enzymes can hydrolyze polyP, the process is slow and moreover, could result in the removal of phosphates from other physiologically important compounds such as adenosine diphosphate. Another approach explored is the use of naked cationic structures such as polyethylenimine (PEI) and polyamidoamine (PAMAM) dendrimers, which bind polyP electrostatically and attenuate thrombosis in murine models[25,27]. However, the concentrations required to provide thromboprotection were shown to be toxic[28–30], precluding their clinical utility. Previous studies from our laboratories also demonstrate the proof-of-concept design of universal heparin reversal agents (UHRA) as polyP inhibitors, although they exhibited some bleeding effects at therapeutic doses[31–34].

Herein, we report an inhibitor design concept based on switchable protonation states in the development of selective polyP inhibitors, termed macromolecular polyanion inhibitors (MPIs). Screening and identification of potent polyP inhibitors is achieved by optimizing the charge density and structure of ligands on a biocompatible polymer scaffold. The selectivity and binding strength are enhanced based on structures that carry the minimal quantity of cationic charge at physiological pH required to bind polyP, and alter their charge state upon binding, to satisfy the positive charge requirement of inhibiting polyP. Lead MPI candidates are identified through a series of biophysical and biological assays. The activities of lead MPI candidates are initially assessed in vitro, and further evaluated for their antithrombotic activity in mice utilizing various thrombosis models. MPI's influence on bleeding and the dose tolerance in vivo are also investigated.

## Results

### Concepts guiding our macromolecular polyanion inhibitor (MPI) design

We generated an MPI library (Table 1) for the initial screening to identify lead inhibitors with selective polyP binding. The concept of the MPI design is shown in Fig. 1. The key property considered was the switchable protonation behavior of weakly basic amines in the designed MPIs[35–37]. The MPIs should possess a low cationic charge state at physiological pH while still being sufficiently charged to initiate binding to polyanionic polyP. Upon binding of polyP to MPI, the changes in the local environment at the polyP-MPI interface enable an increase in the protonation state of MPI, resulting in increased cationic charge and the formation of a highly stable MPI-polyP complex (Fig. 1A, B) thus providing selectivity.

Key principles guiding the structural design of the MPIs explored in this work (Fig. 1) included the identification of a cationic binding group (CBG) displaying significant proton uptake during polyanion binding, as well as the required density, spacing, and shielding of those CBG ligands on a biocompatible scaffold to achieve the desired target binding characteristics, in this case to polyP.

### The selection of ideal CBG structures

Our goal in this study was to identify CBGs that efficiently bind polyP while offering good biocompatibility. Every orthophosphate-subunit in a polyP chain bears one strongly acidic hydrogen ($pK_a = \sim 0-3$), giving this biopolymer an unusually high charge density. To enable high-affinity binding to its cognate binding element on polyP, each unbound CBG ligand must therefore be capable of switching from a low charge state to a higher charge density that improves its ion-pairing with polyP at physiological pH (7.4). In this way, the total quantity of charge on the polycationic MPI is low at physiological pH (7.4), which should improve its biocompatibility and reduce non-specific binding.

Putative CBG chemistries were evaluated on these criteria, with alkyl amines having forms similar to *N,N,N′,N″,N″*-pentamethyldiethylenetriamine (PMDETA) being identified as likely to offer cationic amine residues with $pK_a$ values appropriate for use as charge-switchable CBGs[38]. A comparison of the properties of a number of PMDETA-inspired amine structures then led to the selection of two putative CBG leads (Supplementary Table 1): CBG I (a linear amine with a two-carbon alkyl linker) and CBG II (a linear amine with a three-carbon alkyl linker). Their structures are depicted in Fig. 1E. Each of these putative binding groups has at least one amine residue with $pK_a > 8$, ensuring at least one cation is present per unbound ligand under physiological pH to initiate polyP binding. The remaining amines on each proposed binding group have a reported $pK_a < 7$, making possible a change in their charge state in response to the change in local environment (e.g., lower local dielectric) accompanying binding to polyP. Further tuning of this desired charging-switching effect was sought by including a study of the dependence of the $pK_a$ of adjacent amine residues of the CBG on the length of the alkyl spacer between those amines. It is known that those protonation states depend on amine spacing due to the strong electrostatic repulsion that occurs between proximal positive charges[39].

### CBG ligand display on the dendritic polymer scaffold

In our previous work toward development of polyanion inhibitors[31–34], we reported on a scaffold with a hyperbranched polyglycerol (HPG) core presenting a swollen PEG corona that accounts for ~30 mol% of the copolymer scaffold (HPG-PEG). The PEG corona was shown to shield ligands covalently linked to the HPG core in a manner that minimizes non-specific interactions so as to achieve a high level of biocompatibility[31,34].

We therefore studied the properties and relative performance of our two putative CBGs when presented at different densities and on different HPG-PEG representations in terms of scaffold size. Two distinct HPG-PEG core scaffold sizes of 10 and 20 kDa molecular weight, respectively, were explored. Post functionalization of each HPG-PEG scaffold with different CBGs at different densities then generated a library of MPI candidates for evaluation (Table 1). Candidates within the library were chosen to explore the effects of several variables, most notably CBG chemistry, CBG presentation density, and scaffold molecular weight. Our hypothesis was that the library could serve as a means to elucidate the effects of the CBG and its presentation on the HPG-PEG scaffold on the desired charge-switching properties[40], and how those properties might be tuned to improve polyP binding, biocompatibility, and inhibition of polyP-mediated procoagulant activity in human plasma.

In this study, we included previously identified polycationic polyP inhibitors (UHRA-8 and UHRA-10) to serve as references from which the improved activity and increased biocompatibility of the charge-switching MPI candidates could be measured. UHRAs employ a strong Me$_6$TREN cationic ligand that, when presented on the HPG-PEG scaffold of UHRA carries at least two positive charges in the unbound state at pH 7.4. Its potential for charge-switching is therefore reduced. Detailed characteristics of the MPI candidates within the library,

**Table 1 | Library of MPIs detailing the structural characteristics at pH 7.4 and 25 °C**

| | Polymer scaffold (kDa)[a] | Avg. number of R groups[b] | Avg. number of charges at pH 7.4[b] | $K_d$ (nM)[c] | R group (CBG) structure |
|---|---|---|---|---|---|
| MPI 1 | 21 | 17.7 ± 0.3 | 23 ± 2 | 141 ± 12 | |
| MPI 2 | 23 | 23.5 ± 0.7 | 35 ± 3 | 186 ± 90 | |
| MPI 3 | 21 | 31.2 ± 0.6 | 42 ± 4 | 75 ± 36 | |
| MPI 4 | 23 | 23.1 ± 0.2 | 26 ± 2 | 92 ± 22 | |
| MPI 5 | 21 | 30.2 ± 0.4 | 32 ± 3 | 82 ± 19 | |
| MPI 6 | 10 | 10.9 ± 0.1 | 16 ± 1 | 388 ± 108 | |
| MPI 7 | 10 | 15.5 ± 0.1 | 20 ± 2 | 198 ± 63 | |
| MPI 8 | 10 | 11.7 ± 0.1 | 12 ± 1 | 198 ± 36 | |
| MPI 9 | 10 | 14.2 ± 0.2 | 16 ± 1 | 193 ± 134 | |
| UHRA-8 | 21 | 27.4 ± 0.7 | 45 ± 5 | 3 ± 4 | |
| UHRA-10 | 10 | 14.9 ± 0.2 | 24 ± 3 | 95 ± 52 | |

[a]The molecular weight of polymer scaffold was determined using GPC-MALLS.
[b]The number of binding groups was determined by conductometric titrations and the average number of charges by potentiometry.
[c]The binding affinity presented is towards SC polyP (P110) at physiological conditions (25 °C and pH 7.4).

including their NMR spectra, conductometric titration curves, and GPC profiles are reported in the supplementary data (Supplementary Figs. 1, 2, 12–19), with findings from those characterizations provided below.

**Protonation behavior of MPIs**
We first sought to identify MPIs within our library presenting a relatively low charge density at physiological pH, but adopting a highly charged state upon binding to polyP due to the highly anionic microenvironment surrounding the polyP partner. The two candidate CBGs were chosen with the expectation that only a fraction of their alkylamines will be protonated prior to polyP binding, while then having the ability to support up to three charges per CBG if there is an energetic incentive to adopt this state[36,41].

When free in aqueous solution, CBG I and CBG II carry two amines with a $pK_a > 7.4$ (Table 1, Supplementary Table 2)[39]. It is thought that these two amines are those on the extremities of either CBG structure (Fig. 1C3). The central amine on free CBGI has a $pK_a$ of 4.1, while a lower $pK_a$ is observed for the central amine of CBGII, as expected from the closer amine spacing and potential for stronger charge-charge repulsion[39].

How charge speciation within each of these CBGs changes with a change in each ligand's microenvironment upon attachment to the HPG-PEG scaffold was unknown. Potentiometric titration studies were therefore used to determine the overall average protonation strength of each MPI as a function of pH. The average number of CBGs per MPI was then obtained from conductometric titration, enabling the average charge state of each CBG present on each MPI at physiological pH (Table 1) to be calculated using the Henderson–Hasselbalch equation (Eq.1).

$$pH = pK_a + \log_{10} \frac{[A^-]}{[HA]} \quad (1)$$

Knowing this, stability constants could be regressed from the potentiometric titrations using Hyperquad software (http://www.hyperquad.co.uk/HQ2013.htm). A representative speciation plot and a summary of log K values of different amines for selected MPIs and UHRAs are given in Fig. 2A and Supplementary Table 2. For either CBG, a general reduction in the pKa value of each alkylamine is observed when the CBG is covalently bound on the HPG-PEG scaffold, likely due to a reduction in the local dielectric. As a result, within the scaffold, only one amine on either CBG is fully protonated at pH 7.4, with the

remaining two either partially protonated or deprotonated. The charge density (range from -1 to 2 charges/kDa) of many of the unbound MPIs in the library fall well below that of UHRA (2.4 charges/kDa), and therefore may be sufficiently low under physiological conditions to limit off-target interactions in plasma, especially given the shielding effects provided by the PEG corona. Importantly, significant charging potential remains available to facilitate strong binding to polyP.

**Analysis of MPI binding to polyP using surface plasmon resonance**
We next investigated the polyP binding properties of the MPI candidates using surface plasmon resonance (SPR) to determine the influence of switchable protonation behavior of MPI candidates. The apparent binding affinity of each MPI towards short chain (SC) polyP was determined; the results of which are summarized in Table 1. Binding profiles are shown in Supplementary Fig. 3 for all MPI candidates and the UHRA controls (see also Supplementary Fig. 4). As expected, all MPI candidates exhibited an equilibrium dissociation constant ($K_d$) in the sub-micromolar range in physiological buffer conditions. Across the board, MPIs, based on a lower molecular weight (10 kDa) scaffold (MPI 6−MPI 9) exhibited slightly weaker binding when compared to those based on a 20 kDa core (MPI 1−MPI 5 and UHRA-8). From the perspective of our design concept, the most important finding of this SPR study is that binding strengths similar to those observed for UHRAs can be realized using MPIs with lower charge at physiological pH, presumably via a charge-switching event.

**Analysis of MPI binding to polyP via isothermal titration calorimetry**
To further probe the binding behavior of MPIs to polyP, isothermal titration calorimetry (ITC) analyses were performed using representative MPIs. The dissociation constant ($K_d$) for MPI/polyP (P45) was measured along with the entropy ($\Delta S$), the enthalpy ($\Delta H$), and stoichiometry ($N$) of the binding reaction, giving deeper insight into the driving force of the complexation reaction, are listed in Table 2. Representative ITC thermograms and cumulative binding curves for free CBG I and MPI 3 with polyP are given in Supplementary Fig. 5 and Supplementary Table 3. Compared to free CBG, MPI 3 or MPI 9, exhibits much stronger binding, implying the multivalent presentation of CBG I on MPI 3 or CBG II on MPI 9 greatly increases binding affinity to polyP through avidity effects (Supplementary Fig. 5 and

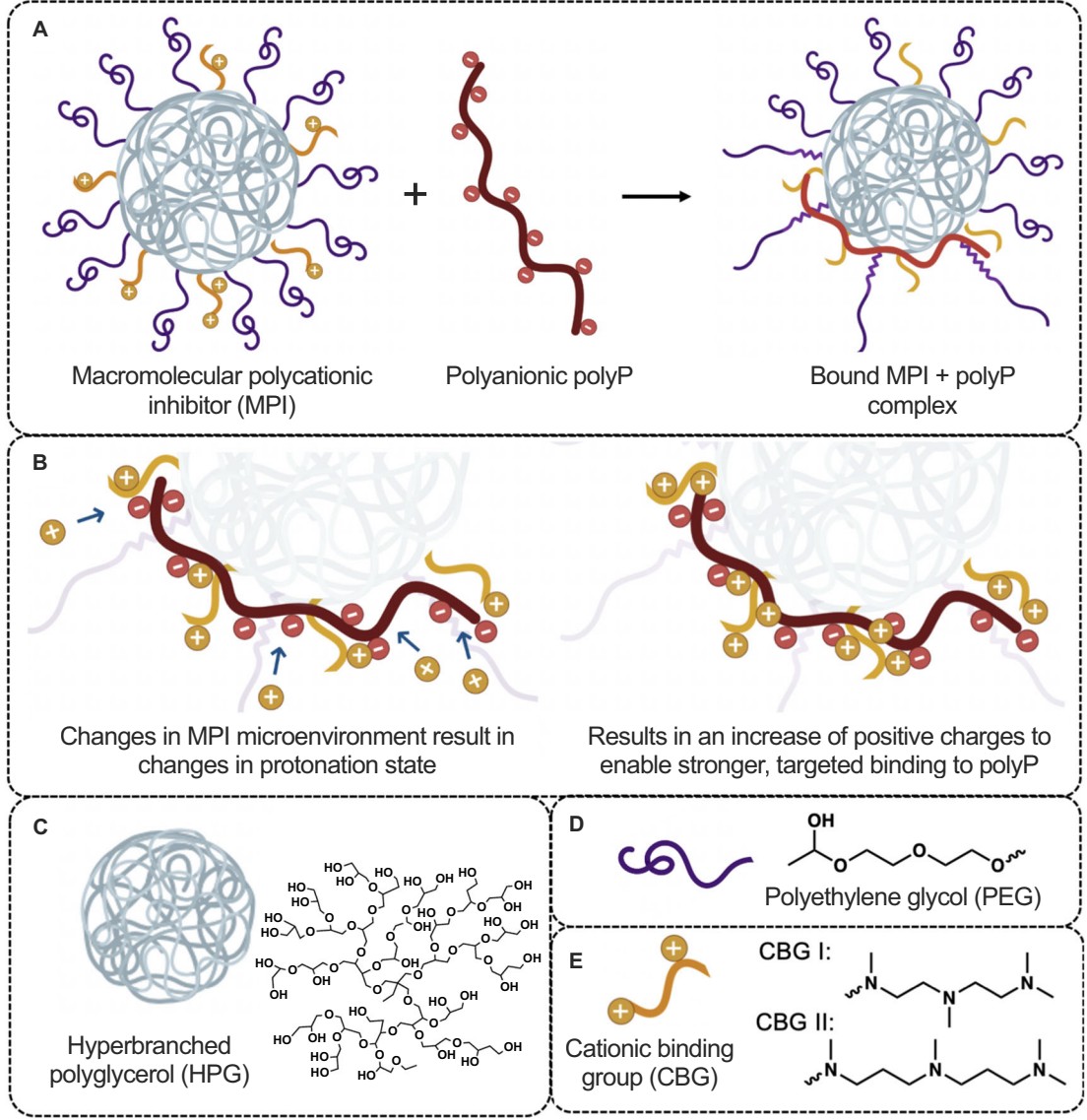

**Fig. 1 | Schematic representation of general macromolecular polycationic inhibitor (MPI) binding to polyP. A** MPI binds polyP to form a stable (MPI + polyP) complex. **B** Zoom-in of charges on MPI and polyP shows that as the cationic charges on MPI initiate binding to the negative charges on polyP, the changes in the electronic microstate of MPI induce a change in the susceptibility of protonation of MPI amines, resulting in a tunable protonation state capable of recruiting protons to successfully bind polyP. The recruitment of protons upon MPI-polyP binding were demonstrated using isothermal titration calorimetry measurements. **C** Structure of hyperbranched polyglycerol (HPG), polymer core in MPI. **D** Structure of polyethylene glycol (PEG), brush structure on MPI. **E** Structures of cationic binding groups CBG I and CBG II conjugated on HPG-PEG surface to bind polyP. The $pK_a$ values of amine nitrogen atoms on CBG I attached to HPG-PEG are 8.4, 7.0, & 3.7 and on CBG II attached to HPG-PEG are 8.9, 6.5 and 3.6 based on potentiometric titrations. This figure was created in part using BioRender.com.

Supplementary Table 3). Regardless of CBG or HPG-PEG scaffold size, the binding of MPIs to polyP is enthalpically driven, with binding affinity compensated by the more ordered state bound macromolecules (entropically opposed binding). These results can be rationalized through consideration of three classic binding effects. First, one must consider classic ion-pairing models, which show that amine (cation)−phosphate (anion) pairing is enthalpically driven. Second, the loss in configurational and translational entropy (ordering) that naturally accompanies macromolecular binding events. Finally, to a lesser degree the Hofmeister effect, since the ionic part of the phosphate group is a weak structure maker.

The strength of the enthalpic driving force for binding shows a dependence on the CBG used. A comparison of polyP binding thermodynamics of MPI 3 (bears CBG I) and MPI 5 (CBG II), which use the same HPG-PEG scaffold and offer a comparable number of CBGs (R groups) per molecule, indicates a stronger enthalpy of binding to the CBG I ligands, but that the corresponding entropy loss is more than offsetting. The more favorable binding enthalpy is likely due, at least in part, to the higher $pK_a$ of the second protonation state of the CBG I ligand in MPI 3 compared to that of CBG II used in MPI 5. The concomitantly greater loss in entropy in the MPI system is almost certainly due in part to standard enthalpy-entropy compensation effects, but the excess entropy loss relative to that expected from entropy-enthalpy compensation would suggest that the formation of strong ion pairs with the more closely spaced amines on the CBG I ligand (two carbon (ethyl) spacer) requires significantly greater structuring of the polyP chain.

**Demonstration of switchable protonation behavior of MPIs**

Isothermal titration calorimetry (ITC) was also used to probe changes in the protonation states of the CBG ligands during polyP binding. The SPR and ITC analyses reported above show that MPIs carry a relatively

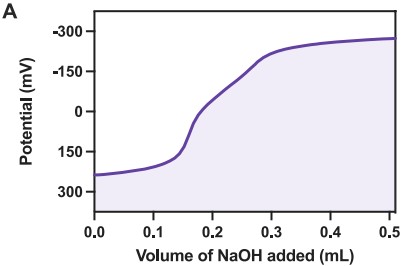
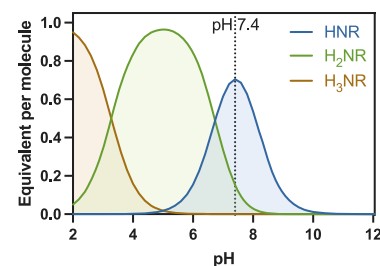
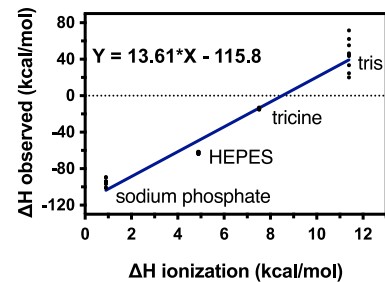

**Fig. 2 | Observed enthalpy in different buffer environments compared to heat of ionization shows recruitment of protons upon MPI binding to polyP.**
**A** Evaluation of free unbound MPI charge content. Potentiometric titrations were used to evaluate the charge content of the MPI library at 25 °C and 160 mM NaCl. A sample titration of 0.15 M NaOH into a solution of acidified MPI 3 while measuring the change in potential is shown. **B** The speciation plot calculated from titration of MPI 3 with NaOH. **C** $\Delta H_{observed}$ obtained from binding MPI 3 and polyP (P75) using ITC200 in four buffers plotted against each buffers' $\Delta H_{ionization}$, at pH 7.4, 150 mM NaCl and 25 °C.

low charge density at physiological pH, yet offer comparable polyP binding affinity to that of the higher charge density UHRA control. We hypothesized that this is due to switchable protonation states associated with CBGs, resulting in a higher cationic charge density of the MPI upon binding to polyP.

Our potentiometric titration analyses showed that either CBG I or CBG II, when presented on an MPI, carries essentially one positive charge per CBG in the unbound state at physiological pH. For either CBG, a second amine is present with a $pK_a$ of 6 – 7 (Supplementary Table 2, Fig. 2A, B), making possible facile recruitment of a second cation to facilitate strong binding to the negatively charged polyP polyanion. If this proposed charge-switching occurs, it should be possible to estimate the degree of proton recruitment using ITC.

To verify this switchable protonation ability of the MPIs generated, a series of ITC experiments using a representative candidate, MPI 3, on its binding to P75 were conducted in several buffers offering a range of different known heats of ionization $\Delta H^0_{ion}$ (Eq. 2). In this well-established method[42,43], recruitment of protons in the binding process can be observed through the effect $\Delta H^0_{ion}$ has on the measured binding enthalpy $\Delta H^0_{exp}$. Therefore, $\Delta H^0_{exp}$ was measured in buffers of differing $\Delta H^0_{ion}$ and plotted against the $\Delta H^0_{ion}$ of each buffer[42,43] (Fig. 2C). Using Eq. 2, we are able to determine if protons are recruited by MPI 3 during P75 binding by computing the slope of the resulting plot.

$$\Delta H^0_{exp} = \Delta H^0_{bind} + n\Delta H^0_{ion,} \qquad (2)$$

In a less complex binding process, such as a small-ligand to protein binding event, this slope can be directly correlated to the number of protons recruited in the binding process[44]. In the case of the MPI-polyP binding process, however, the dispersity of both polymeric

binding partners makes it difficult to extract an exact value. The line of best fit (Fig. 2C) indicates that -13.6 protons are recruited when MPI 3 binds to P75, but the errors on this measurement give a 95% confidence interval on this slope from 7.0 to 20.3 protons recruited per binding event. Unbound, MPI 3 carries ~31 CBGs and a total charge of 42 charges per MPI at pH 7.4. The recruitment of 14 ± 7 protons therefore represents a significant increase in the number of positive charges—roughly a 32% increase in cationic charge upon binding. As a result, the charge density of MPI 3 is raised to ~2.6 ± 0.6 charges/kDa. This significant increase in the cationic charge density of the MPI is consistent with our molecular design in which charge-switching enables a bound MPI to exhibit significantly higher charge density than present in its unbound state at physiological pH.

## Screening and identification of polyP inhibition in human plasma by MPIs

As a next step, the biological activities of the MPI candidates were investigated by determining their polyP inhibition activity using plasma clotting and thrombin generation assays in human plasma. Two sizes of polyP, long chain (LC) polyP, and short chain (SC) polyP were used. UHRA-8 and UHRA-10 were used as benchmarks.

In the initial study, plasma clotting triggered by recalcification in the presence of LC polyP (P700) was used to investigate the inhibitor activity of MPI candidates. The ability of MPIs to inhibit the procoagulant effects of polyP is shown in Supplementary Fig. 6. As can be seen, plasma without polyP (i.e., clotting initiated by calcium only) resulted in an average clot time of 166 ± 3 s, while the control with LC polyP had an average clot time of 117 ± 1 s, demonstrating the procoagulant activity of LC polyP. Increasing concentrations of MPIs inhibited the procoagulant activity of polyP, as evidenced by the normalization of the clotting time in comparison to the buffer control. Some MPIs, e.g., MPI 5 and MPI 9, normalized the clot time to that of buffer at concentrations as low as 12.5 μg/mL. Other candidates such as MPI 1, MPI 2, and MPI 3 showed slightly lower activity at the same concentration. On the other hand, UHRA-8 demonstrated a considerable prolongation of clot time at higher concentrations, which suggests off-target inhibition of the clotting cascade independent of polyP, with clot times >550 s. However, MPI candidates did not show this dramatic increase even at high concentrations (Supplementary Fig. 6).

Next, we investigated thrombin generation in the presence of LC and SC polyP using calibrated automated thrombography (TGA). In this approach, a calibrated fluorogenic substrate-cleavage assay was used to measure the quantity of thrombin generated. The addition of LC polyP, a potent activator of clotting, significantly shortened the clot time of plasma, while the addition of MPI normalized the thrombin generation curve (Fig. 3A–F). Clotting parameters, including lag time, endogenous thrombin potential (the total amount of thrombin

**Table 2 | Summary of ITC data characterizing the thermodynamic properties, stoichiometry (N) and equilibrium dissociation constant $K_d$ for binding of MPI to polyP at pH 7.4 and 25 °C**

| polyP[a] | MPI[b] | N[c] | $K_d$ (μM) | ΔG (kcal/mol) | ΔH (kcal/mol) | TΔS (kcal/mol) |
|---|---|---|---|---|---|---|
| P45[d] | MPI 3 | 0.7 | 0.737 ± 0.01 | −8.37 ± 0.04 | −107.0 ± 0.8 | −98.7 ± 0.8 |
| | MPI 5 | 0.8 | 0.95 ± 0.05 | −8.22 ± 0.02 | −75.0 ± 0.4 | −66.8 ± 0.4 |
| | MPI 7 | 1.0 | 1.27 ± 0.09 | −8.05 ± 0.06 | −37 ± 1 | −29.0 ± 0.9 |
| | MPI 9 | 1.1 | 1.2 ± 0.2 | −8.08 ± 0.02 | −50.9 ± 0.5 | −42.9 ± 0.5 |

[a]Used in the ITC cell.
[b]Added to cell via syringe.
[c]Ratio of MPI to polyP.
[d]Buffer used was sodium phosphate buffer composed of dibasic phosphate buffer (Na$_2$HPO$_4$), monobasic phosphate buffer (NaH$_2$PO$_4$) and NaCl. NaCl concentration is 10 mM.

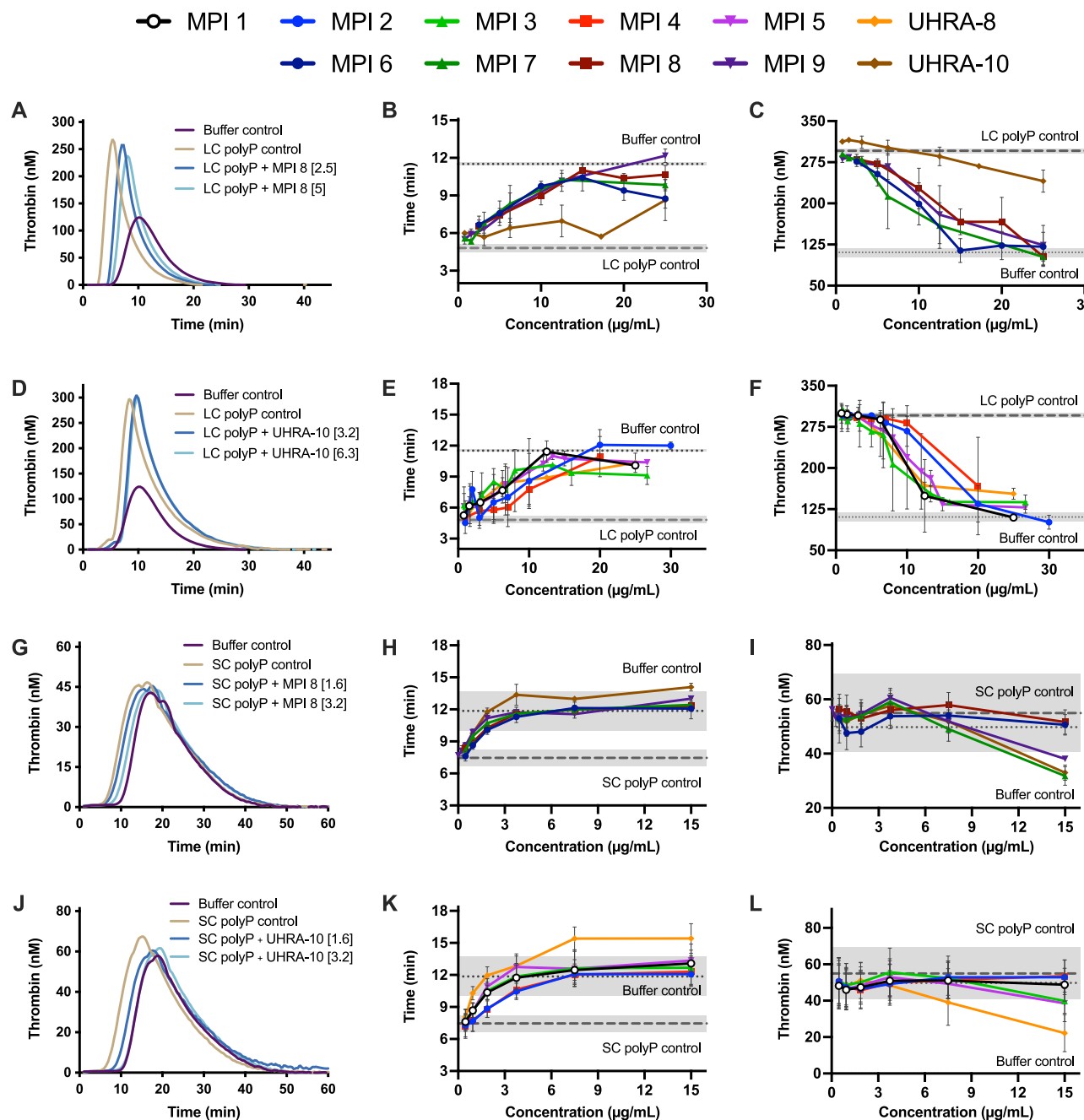

**Fig. 3 | MPI library inhibits both bacterial-sized polyP and platelet-sized polyP procoagulant activity in vitro. A–C** inhibition of LC polyP by MPI 6–MPI 9. **D–F** inhibition of LC polyP by MPI 1–MPI 5. **G–I** inhibition of SC polyP by MPI 6–MPI 9. **J–L** inhibition of LC polyP by MPI 1–MPI 5. LC polyP assays were performed with 20 μM LC polyP, while SC polyP assays were performed with 8.3 fM TF (tissue factor) and 5 μM SC polyP. Column 1 panels represent thrombin generation curves with a smoothing factor of 2 from 1 biological replicate as a representation. Column 2 panels represent lag times from thrombin generation assays. Column 3 panels represent peak thrombin from thrombin generation assays. Columns 2 and 3 represent mean values +/− SD from $n = 3$ biologically independent experiments.

generated, quantified as the area under the curve), time to peak and peak thrombin were evaluated in these experiments. The dose-dependent response of polyP inhibition activity was notable in the case of some MPI candidates (MPIs 1, 6, and 8) for which an almost complete return of the thrombin generation parameters to that of the buffer control was observed. Together, these data demonstrate the inhibitory activity of MPIs against LC polyP.

Next, we investigated the ability of MPI candidates to inhibit SC polyP. While SC polyP is a less potent activator of the contact pathway, it plays many roles in the downstream events of clotting and remains an important target in the development of an antithrombotic. Less

pronounced effects were seen than in the case of LC polyP in thrombin generation (Fig. 3G, J). A titration of the MPI candidates revealed a dose-dependent response of short chain polyP inhibition, normalizing TGA parameters to that of the buffer control (Fig. 3G–L). MPI candidates 1, 6, and 8 resulted in the normalization of lag time, endogenous thrombin potential, peak thrombin, and time to peak thrombin, i.e., to values close to those observed with the buffer control.

From the dose-response curves generated for each MPI candidate, we calculated the half maximal inhibitory concentration ($IC_{50}$) as listed in Supplementary Table 4. While MPI candidates and UHRA-10 generated classical sigmoidal dose-response curves which could be fitted

using the Hill equation, UHRA-8 did not fit this model. Since LC polyP is a potent activator of the contact pathway, the lag time was taken as the key parameter affected by the addition of LC polyP and was used to generate the $IC_{50}$ of each MPI. While a range of $IC_{50}$ values towards SC and LC polyP were observed for each MPI and UHRAs, some general trends could be extracted from these data. Most notably, all 20 kDa and 10 kDa MPI candidates demonstrated a lower $IC_{50}$ value towards LC polyP than did UHRA-8 or UHRA-10, respectively (Supplementary Table 4). Furthermore, although MPI candidates possess significantly less charge than their UHRA counterpart at neutral pH (about 30–50% lower), no significant loss of inhibition efficacy against either LC or SC polyP was observed.

### Hemocompatibility of MPI candidates

After confirming the polyP inhibitory activity of the MPI candidates, we investigated their blood compatibility as a metric for further selection for in vivo studies, as polycations are known to adversely interact with blood components[29,30]. Initial experiments to screen the MPI library used a tissue factor (TF)-initiated plasma clotting analysis in FXII-deficient plasma in the absence of polyP. Results are shown in Fig. 4A. In all cases except UHRA-8, there was no significant difference between the clot time with any of the MPIs and the buffer control, implying that MPI did not cause any significant changes to clot time in this TF-triggered clotting system. Improvements over UHRA-8, however, highlight the increased biocompatibility of these MPI candidates.

We further utilized thrombin generation in TF-initiated plasma clotting. Various parameters including lag time and the amount of thrombin generated were determined (Fig. 4B, C). As shown, there are considerable improvements associated with MPI candidates on various TGA parameters compared to previous polyP inhibitors (UHRA-10 and UHRA-8) with increasing concentration. While most MPI candidates do not cause much variation in TGA parameters compared to the buffer control at low concentrations (10–20 µg/mL), some candidates showed large changes in thrombin generation at high concentrations (most notably MPI 3 and MPI 9). Select MPI candidates (MPI 1, MPI 2, MPI 4, MPI 6, and MPI 8) demonstrated no effect on TGA parameters for TF-initiated clotting reactions, suggesting that they do not interfere with normal hemostasis and have high biocompatibility.

### Selection of lead MPI candidates for further studies

The in vitro screening studies conducted thus far, generated data that helped to identify lead candidates for further studies. An ideal MPI should show minimal deviation in terms of its influence on blood coagulation in the absence of polyP compared to the buffer control, while maximizing the polyP inhibition activity. Based on the TGA data, some MPI candidates induced large changes in TF-initiated thrombin generation at high concentrations (most notably MPI 3 and MPI 9), while MPI 1, MPI 2, MPI 4, MPI 6, MPI 8, did not trigger such changes, suggesting minimal side effects on hemostasis.

Based on the preceding finding and the polyP inhibition data (Fig. 3, Supplementary Table 4), three MPI candidates (MPI 1, MPI 6, and MPI 8) stood out as the strongest candidates for further study. Each of these lack unwanted effects on blood clotting parameters in the absence of polyP, while exhibiting strong inhibition of polyP activity at low concentrations. These lead MPI candidates were further characterized to investigate their compatibility in more complex and sensitive systems. Measurements were performed to confirm that the lead MPI candidates do not exhibit incompatibility with blood components that might preclude them from therapeutic applications. These studies also confirm the significant advantage of the polyP inhibitor design concept, which is enabled by their switchable protonation behavior and minimal charge present at physiological conditions.

### Influence of MPIs on whole blood coagulation and fibrin clot morphology

We investigated the influence of lead MPI candidates in whole blood clotting and fibrin clot formation. Rotational thromboelastometry (ROTEM) measurements were used for evaluating the lead candidates as conditions used in these experiments are considered to more closely mimic the in vivo situation than other coagulation assays[45]. A representative whole blood ROTEM profile of MPI 8 is shown in Fig. 4D. Parameters including clot time and maximum clot strength are shown in Fig. 4E, F. Unlike UHRA-10 and UHRA-8, the lead MPI candidates did not change the clot time or clot strength in comparison to the buffer control. Overall, the ROTEM profiles were closer to the condition in which buffer is added to whole blood. These results demonstrate that the lead MPI candidates do not affect blood coagulation nor interfere with hemostasis.

Next, the influence of MPI on final clot structure was investigated as a function of polyP. Clot structure was visualized using scanning electron microscopy (SEM). Only MPI 8 was used for these experiments. Clot morphology and fibrin dimensions were calculated and are shown in Fig. 4G, H. The overall clot morphology and microstructure in the presence of MPI 8 were similar to the buffer control, as were the fibrin diameters. The addition of polyP increased the fibrin diameter as expected[15]. Clot morphology and fibrin diameters were normalized to that of the buffer control when polyP was inhibited by MPI 8.

### Antithrombotic activity of lead MPI candidates in vivo

The antithrombotic activities of the lead MPI candidates, MPI 1, MPI 6, and MPI 8, were evaluated in a mouse model of laser-induced cremaster arteriole thrombosis using intravital microscopy (Fig. 5A). MPIs were injected and platelet accumulation and fibrin deposition were dynamically measured using fluorescent antibodies to allow direct observation of clot formation following laser injury (Fig. 5D). Based on results obtained from an average of 8 injuries, the rate of accumulation of platelets and fibrin deposited were derived from the fluorescence intensities. These parameters were used as an indication of thrombus growth rate. A reduction in both platelet accumulation rate and the total amount of platelets accumulated were observed for mice treated with 100 mg/kg MPI 1, MPI 6, and MPI 8 compared to mice that received only the saline control (Fig. 5B, C). In the case of mice treated with MPI 8, a reduction in fibrin accumulation following the injuries was also observed. Thus, MPI 8 was selected for further studies.

The ability of MPI 8 to inhibit thrombosis in a mouse model of $FeCl_3$-induced carotid artery injury (Fig. 5E) was examined. Along with MPI 8, a previous generation polyP inhibitor (UHRA-10) was used as a control. As shown in Fig. 5F, a dose of 100 mg/kg of MPI 8 was more effective in preventing occlusion, whereas UHRA-10 did not prevent occlusion under identical dosage conditions. MPI 8 at this concentration was more effective at delaying the patency as compared to a dose of 100 mg/kg UHRA-10. Similar observations were found at higher doses (200 and 300 mg/kg) (Fig. 5G and Supplementary Fig. 7; maximum patency was achieved in this model by these inhibitors). Since polyP is an accelerant to blood coagulation but is not required for clotting, it is likely that 100% patency may not be achieved with polyP inhibitors in this model, consistent with previous observations[26].

We further investigated the efficacy of MPI 8 in the inferior vena cava (IVC) stenosis (partial) ligation model of thrombosis at 48 h post-VT (Fig. 5H)[46,47]. Thrombosis was achieved in 8 of 10 mice for each group and these 8 values were used for comparison. MPI 8 treatment yielded significantly smaller thrombi than control treatment as measured by thrombus weight $p = 0.0014$, and thrombus weight/length $p = 0.0201$ (Fig. 5I, Supplementary Figs. 8 and 9).

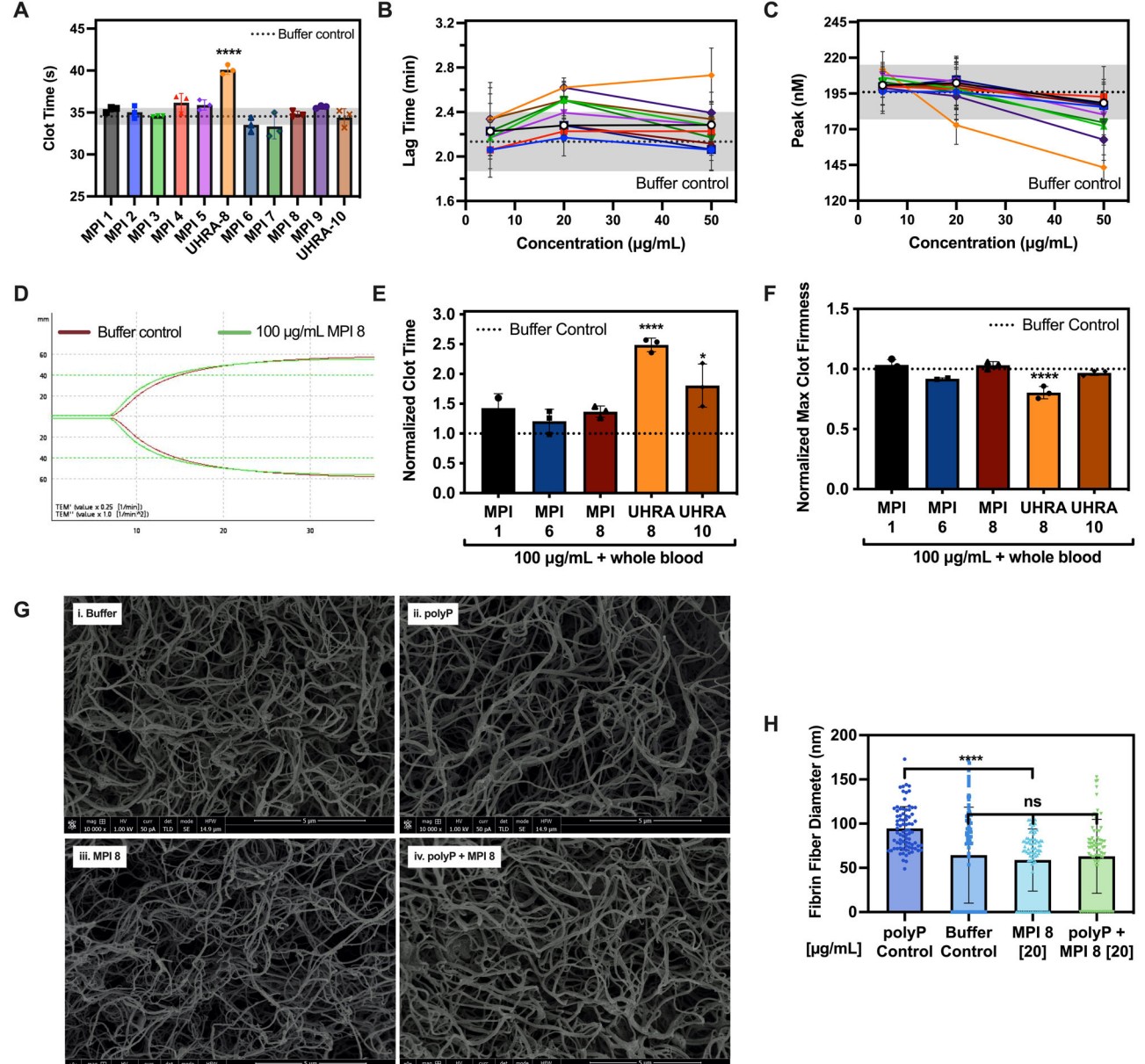

**Fig. 4 | MPI candidates show superior hemocompatibility. A** MPIs do not affect lag time in TF-initiated plasma clotting system. Clot times measured on a coagulometer in a FXII-deficient plasma clotting system with clotting initiated with 100 pM TF. Negative control was a mixture of plasma, HBS and TF without MPI added, shown as grayed area with standard deviation as gray dotted lines. Clot times of plasma with added MPI [100 μg/mL] show no significant difference in clot time relative to the negative buffer control; almost all lie within the standard deviation of the buffer control. Only UHRA-8 was significant with ****$P < 0.0001$. **B, C** Lead MPI candidates show no influence on thrombin generation in plasma. TF initiated human plasma clotting parameters in a thrombin generation assay. Buffer control represents plasma clotted with buffer. MPI, when added in lieu of buffer, show minimal effects on thrombin generation compared to UHRA-8. **B** Lag time. **C** Peak thrombin. **D–F** Lead MPI candidates do not influence whole blood clotting. Rotational thromboelastometry (ROTEM) of fresh citrated human whole blood clotting

activated with $CaCl_2$. 100 μg/mL MPI was added to whole blood and buffer control was performed with each experiment, per donor. **$P < 0.001$, ***$P < 0.0005$, ****$P < 0.0001$. **D** Thromboelastrogram. **E** Clot time normalized to buffer control ($636 \pm 156$ s). ****$P < 0.0001$. **F** Maximum clot firmness normalized to buffer control ($53 \pm 4$ mm). A1-A3 and B2-B3 represent mean values +/– SD from $n = 3$ donors. ****$P < 0.0001$. **G–H** MPI 8 does not interfere with fibrin clot fiber thickness and morphology. Images were acquired using a Helios 650 focused ion beam scanning electron microscope. Scanning electron micrographs of fibrin clots formed in the presence of (**G**) Clot images were taken at three magnifications ×5000, ×10,000 and ×25,000. Images from ×10,000 are depicted. **H** Quantified fibrin fiber diameter. Results represent mean values +/– SD from $n = 2$ independent clots per group. MPI 8 normalized the polyP induced increase in fibrin diameter to that of buffer control. Results analyzed using ordinary one-way ANOVA, two-tailed, with Tukey's multiple comparisons test. ****$P < 0.0001$, ns not significant.

## Safety of the MPI candidates

The effect of the lead MPI candidates on hemostasis was assessed in a mouse-tail bleeding model (Fig. 6A). Mice were treated with MPI 8, saline or unfractionated heparin (UFH) (as a positive control). Bleeding times and hemoglobin loss were recorded after a tail clip. As shown in Fig. 6B, C, doses of MPI 8 of 300 mg/kg resulted in minimal effects on

the bleeding time and hemoglobin loss relative to the saline control. UFH caused a significant prolongation in the bleeding time with an increase in hemoglobin loss. Even at very high doses, MPI 8 showed no signs of prolonged bleeding compared to the saline control. To further probe the effect of MPI 8 on hemostasis, the saphenous vein hemostasis model in mice was used to assess hemostatic clot formation at the

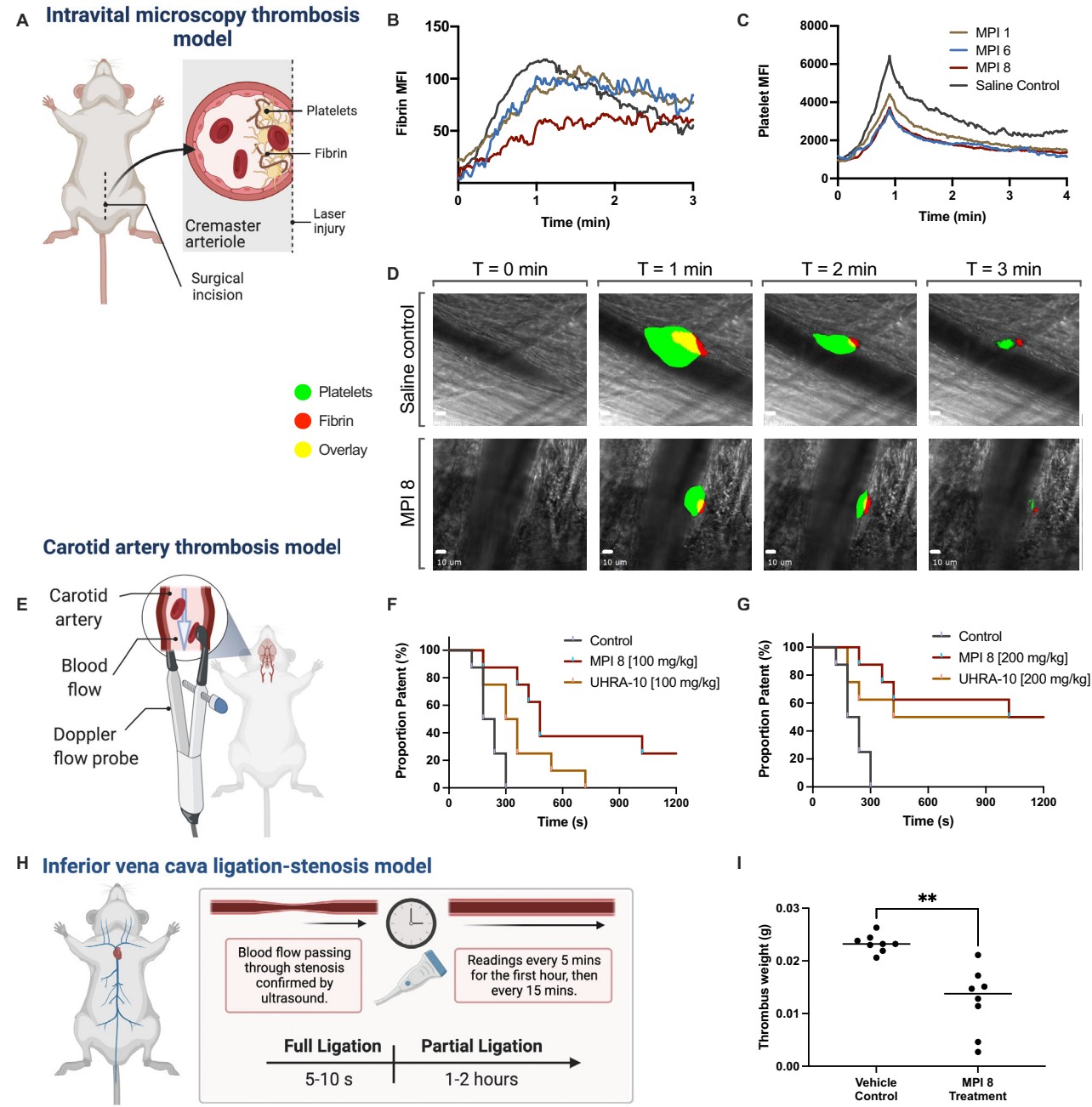

**Fig. 5 | MPI 8 exhibits antithrombotic properties in mice. A–D** MPI 8 reduced fibrin clot formation and platelet accumulation in mouse cremaster arteriole thrombosis model. **A** Schematic representation of the thrombosis model: Platelets and fibrin are tagged with fluorescent antibodies and can be visualized as they accumulate at the site of injury upon laser-induced injury. **B** Median fluorescence intensity representative of fibrin accumulation over time. **C** Median fluorescence intensity representative of platelet accumulation over time. **D** Representative images of thrombus growth at 0–3 mins in mice which were treated with saline or MPI 8. All results shown are mean +/− SD of $n = 3$ mice. **E–G** MPI 8 delays time to occlusion in carotid artery thrombosis model. **E** Schematic representation of the thrombosis model. Artery patency was monitored by Doppler flow probe. Injury was induced by topical application of $FeCl_3$ and patency is plotted versus time, comparing the saline control and MPI 8. UHRA-10 was used as a control. **F** At

100 mg/kg, MPI 8 is more effective at delaying time to occlusion. A log rank analysis test shows that the curve of MPI 8 is significantly different from the curve of UHRA-10 (MPI 8 vs UHRA-10 ***$P < 0.0005$). **G** At 200 mg/kg, MPI 8 and UHRA-10 have reached a similar level of patency, likely a maximum in this model using these inhibitors. All results shown are mean +/− SD of $n = 8$ mice. A log rank analysis test indicates the two curves of MPI 8 compared to the saline control are significantly different (**$P < 0.005$). **H, I** MPI 8 treatment inhibits thrombus formation in inferior vena cava thrombosis model. **H** a representation of the mouse model. **I** Thrombus weights of untreated (vehicle control) and treated (MPI 8) mice. After removal of 2 outliers, due to a lack of clot formation, data were statistically significantly different at $p = 0.0003$ in an unpaired, two-tailed $T$-test with Welch's correction. Results shown are mean of $n = 8$ mice. Figure 5A, E, H were created using BioRender.com.

site of vascular injury following laser-induced rupture of the saphenous vein wall under intravital microscopy[48,49]. After the initial injury in vehicle-treated WT mice and MPI 8 treated WT mice, platelets immediately began adhering to the site of vascular injury to form a visible platelet-rich clot (Fig. 6D, E). Fibrin also began forming around the

platelets at the site of vascular injury. The hemostatic response to vascular injury and clot stability increased after each subsequent injury and continued to grow (Fig. 6F). Platelet fibrin hemostatic clot formation in MPI 8 treated mice (200 mg/kg) was similar to WT control with no detectable decrease in fibrin formation and platelet

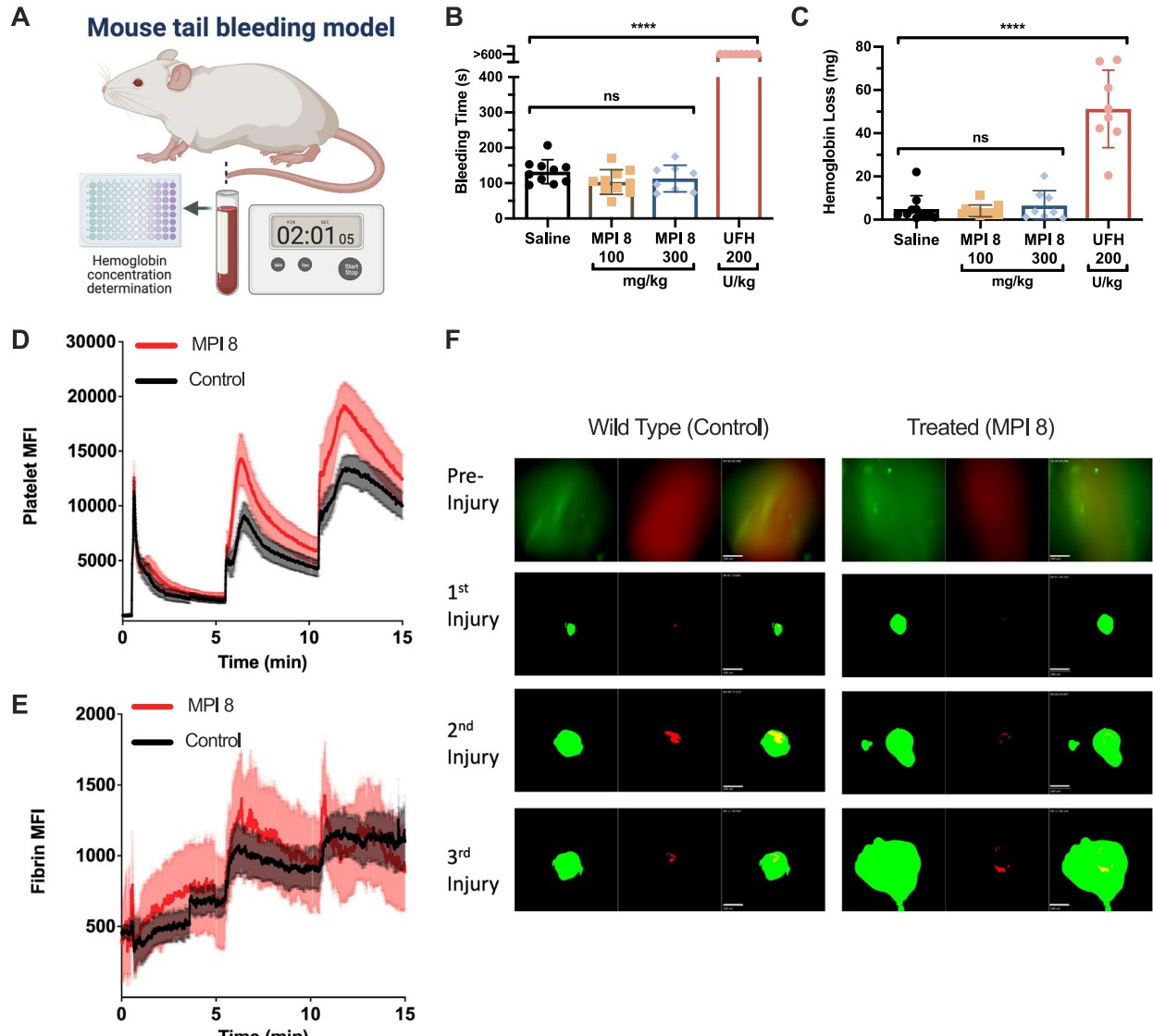

**Fig. 6 | MPI 8 does not induce bleeding in mice. A–C** High doses of the lead MPI candidate do not cause bleeding in mice. **A** Schematic representation of mouse tail bleeding model with C57/BL6 mice. **B** Recorded bleeding time. Mice were injected in a blinded study with up to 300 mg/kg of lead MPI candidates and demonstrated no increase in bleeding side effect, in contrast to mice administered 200 U/kg of unfractionated heparin. **C** Hemoglobin lost by mice. **D–F** MPI 8 did not decrease platelet recruitment and fibrin formation in hemostatic clot formation assessed via saphenous vein hemostasis model. Quantitative analysis of the dynamics of platelet accumulation (**D**) and fibrin formation (**E**) in response to vascular injury in saphenous vein. The kinetic curves represent the mean fluorescence intensity, and the shaded regions are representative of the standard error (SEM). **F** Representative images of platelet (green) and fibrin (red) hemostatic clot formation in response to a repetitive vascular injury of the saphenous vein. Mice were injected with 200 mg/ kg of the lead MPI 8 candidate. The scale bar is 100 μm shown in the left lower corner of the composite image of panel 3 (third from left) for wild type (control) and treated (MPI 8) groups. All results represent mean values +/− SD of $n = 8$ mice per group. Results analyzed using ordinary one-way ANOVA, two-tailed, with Tukey's multiple comparisons test. ****$P < 0.0001$, ns not significant. Figure 6A was created with using BioRender.com.

accumulation as analyzed by the dynamics of platelet MFI and fibrin MFI through repetitive injury. These data suggest that this polyP inhibitor is safe from a hemostatic point of view. It is equally important to highlight that in this regard, MPI 8 is significantly more effective than the best previous generation candidate, UHRA-10.

Next, we investigated the dose tolerance of MPI 8. A single escalating dose-tolerance study in mice was initially performed to test for acute toxicity. Mice were injected with MPI 8 intravenously (i.v.) at high doses (250 and 500 mg/kg) and euthanized after 24 h. Serum levels of lactate dehydrogenase (LDH), aspartate aminotransferase (AST) and alanine aminotransferase (ALT) were used to assess acute toxicity. As shown in Fig. 7A–C, no significant changes were observed in comparison to control mice injected with saline.

Thus, MPI 8 does not elicit acute tissue or liver toxicity under these experimental conditions.

The effects of long-term exposure to MPI 8 in mice was evaluated using an escalating dose injection study (100 to 500 mg/kg) over the course of 14 days (Fig. 7D). MPI 8-injected mice gained weight similar to control mice (Fig. 7E). Serum levels of LDH at termination (Fig. 7F) were not different as compared to controls. The major organs were analyzed histologically at the end of the study. Tissue sections were stained using hematoxylin and eosin (H&E) and examined for tissue damage. Representative images for the 500 mg/kg dose are shown in Fig. 7G and no abnormalities were observed in the lungs, heart, kidneys or liver. Overall, the data show that MPI 8 is well tolerated even at 500 mg/kg.

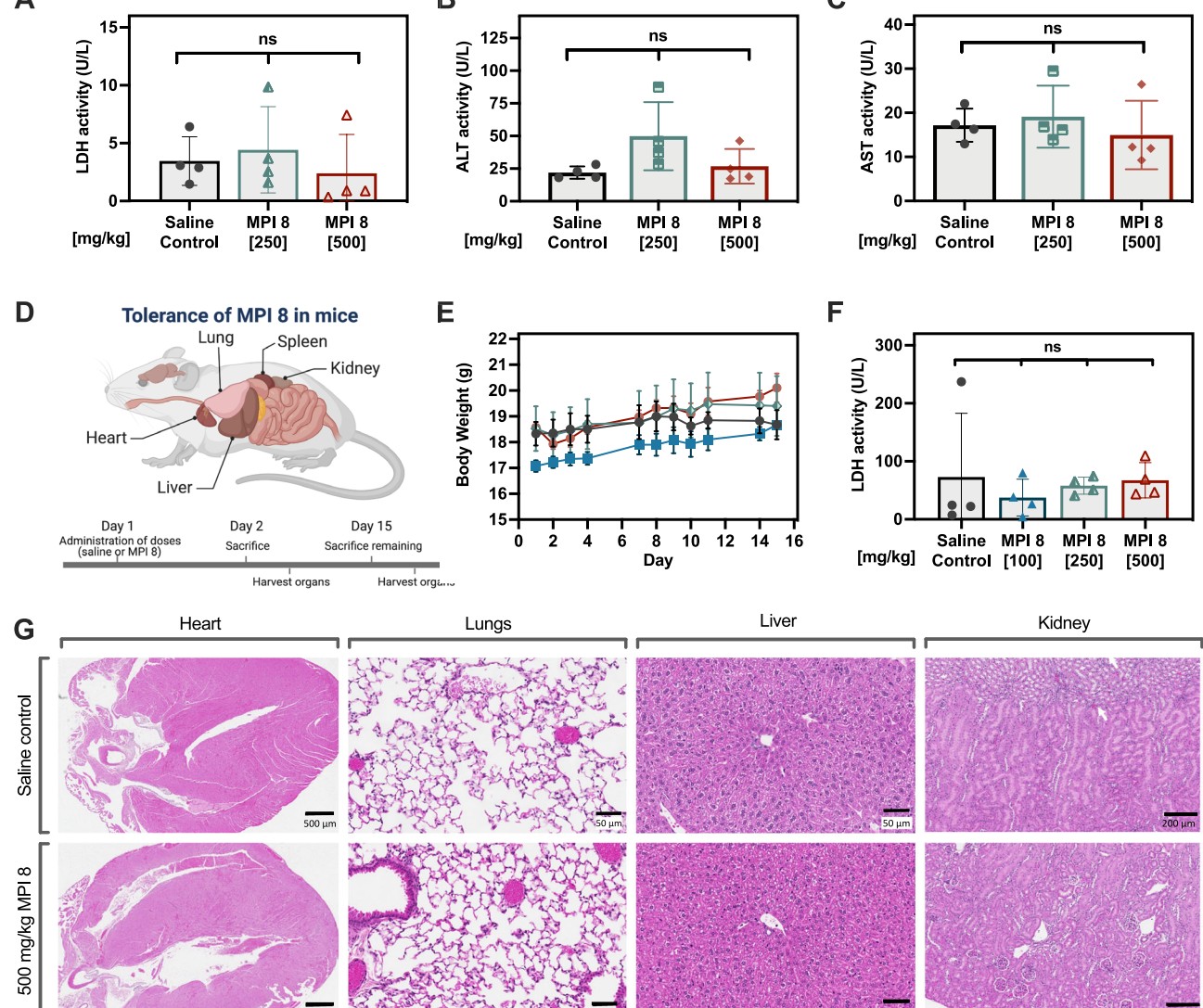

**Fig. 7 | Tolerance of MPI 8 in mice. A–C** MPI 8 were well tolerated in mice at high doses. **A–C** LDH activity, ALT activity, AST activity in mice, respectively, after injecting MPI 8 intravenously. Mice were sacrificed after 24 h (acute study). **D** Schematic representation of mouse chronic toxicity model. Female BALB/c mice were administered either saline or escalating doses of MPI 8, up to 500 mg/kg. Mice were monitored daily and body weights were measured. After 15 days, serum was collected from sacrificed mice and analyzed for LDH levels. Mice injected with MPI 8 showed no significant change in body weight compared to mice injected with saline and no increase in LDH levels. **E** Change in body weight over 15 days. **F** LDH activity. **G** Representative stained (H&E) images of organs collected from mice injected (i.v.) with 500 mg/kg MPI 8. No abnormalities were seen in the heart, lungs, liver and kidneys of mice administered with 500 mg/kg MPI 8 compared to the saline control. All results (**A–C, E–F, G**) represent mean values +/− SD of $n = 4$ mice per group. Figure 7D was created using BioRender.com.

## Discussion

Current anticoagulants target key factors in the coagulation cascade and carry significant risk of bleeding[50–52]. Indeed, to date, there is no effective antithrombotic drug available that is not complicated by a bleeding risk. Recent investigations on FXIa inhibition and FXIIa inhibitors are promising but not yet approved for clinical use[9,53]. Research over the last decade revealed that procoagulant polyanions including polyP are valid targets toward the development of safer antithrombotics[5,6,54–56]. By targeting an optional coagulation initiator and accelerant such as the polyanionic polyP, thrombosis can be attenuated with minimal bleeding risk, leading to a safer antithrombotic[22]. Currently, there are no available therapeutics which successfully target and inhibit polyP. We presented a therapeutic design optimized for targeted inhibition of polyP via specific tuning of weakly basic binding groups paired with an optimized biocompatible polymer scaffold. Our data demonstrated that in mouse models, the lead compound is

well-tolerated and can prevent thrombosis without increased bleeding risk.

While the ability of MPIs to directly bind polyP and inhibit its unwanted procoagulant activity is key to their efficacy, it was also necessary that they retain high biocompatibility to prevent side effects. This requirement necessitated careful balance of the effects of MPIs on hemostasis and thrombosis. As with many other blood components, endothelial surfaces and biomolecules are negatively charged, so carefully controlling the off-target electrostatic interactions is a major challenge in developing such polycationic inhibitors for polyP. In fact, eliminating polycationic toxicity remains a sought-after objective. Thus, we introduced a switchable protonation approach to identify polyP inhibitors with high selectivity. Previous studies by Vacca, Martell and Lehn laid the ground work and highlighted the importance of fine tuning the pKa of polyamines and the changes in pKa of amines with the microenvironment to understand their binding to anionic partners[57–59]. Utilization of such concepts lead

to the development of polycationic structures that target tumors[60], lipid nanoparticle-siRNA delivery systems[61,62]; however, such systems require an external trigger (chemical or physical) in order to fulfill their task. Here, we present a design concept for a stealth cationic inhibitor where the trigger for increasing the cationic charge is the target binding site itself, i.e., polyP.

Using a highly adaptable design approach, we created a class of polyP inhibitors that offer significantly enhanced biocompatibility and strong inhibition of polyP procoagulant activity. These improvements are achieved utilizing ligands bearing multiple alkylamines with at least one protonation state, when the ligand is displayed on a biocompatible HPG-PEG polymeric scaffold, designed to change during target binding. Putative cationic binding group structures described in the literature were screened for their amine $pK_a$ values, charge spacing, and linker chemistry, which may affect their flexibility. Embedding such charge switching cationic binding groups within a biocompatible scaffold known to prevent non-specific interactions resulted in a library of polycations with low charge density in their unbound state, particularly as compared to other polycation inhibitors of polyP[25,32,33,63]. When compared to larger polyanionic macromolecules with lower charge density, the unique features of the MPI's scaffold serve to improve biocompatibility, presumably (at least in part) by limiting nonspecific interactions through the steric repulsion created by the PEG corona (brush layer) on the surface of the MPI. Importantly, the design platform and underlying synthesis chemistry are flexible, potentially enabling specific tuning of the MPI's properties and polyP inhibition efficacy through, for instance, the facile synthesis of a library of compounds of varying scaffold size, CBG structure, and quantity and density of charge and charge-switching potential. Measurements using SPR and ITC showed that the charge-switching MPIs generated strong binding to polyP. Though the unbound MPIs carry significantly less charge density than previous generation UHRA-based polyP inhibitors (ca. 50–85 % of UHRA type)[32], they bind polyP with an affinity similar that of UHRA. All measured binding affinities were in the sub-micromolar range.

Evidence for the realization of this design concept comes from studies with our lead candidate, MPI 8, which show selective polyP neutralizing activity without bleeding risk and with a superior safety profile in multiple animal models. The lead, MPI 8, has a charge density one-half that of the previous best polyP inhibitor, UHRA-10. The lower charge density and its switchable protonation behavior led to a polyP inhibitor with high selectivity, devoid of non-specific interactions with blood components, and with a larger therapeutic window. Library screening approaches allowed us to identify a lead candidate and the data support that MPI 8 does not alter TF-initiated clotting, thrombin generation or clot structure. This lead MPI candidate showed no interference with blood components such as platelets, nor did it alter whole blood clotting. MPI 8 showed good activity for both long chain and short chain polyPs, thus presenting the significant advantage of being able to target multiple clinically relevant sizes of polyP. In addition, MPI 8 does not prevent clotting activation by other negatively charged molecules such as nucleic acids (Supplemental Fig. 10) and also does not change the activity of small phosphate containing biomolecules such as adenosine diphosphate (ADP) (Supplemental Fig. 11). Together these data suggest that MPI 8 has good selectivity to polyPs. Notably, MPI 8 was also able to reverse the polyP effect on clot structure without altering the final clot structure (relative to the condition where no polyP was added), a strong indication that MPI may be able to reverse polyP activity without adversely affecting the final clot.

The antithrombotic activities of MPIs were evaluated in three mouse models of thrombosis. In addition, mouse models were used to determine whether the lead MPIs caused toxicity or excess bleeding. MPI 8 is highly effective in preventing thrombosis without bleeding risk. Other previously investigated polycations such as PAMAM dendrimer, PEI and polymyxin B were shown to be much less effective in these models and these polycations were also quite toxic[25,30]. MPI 8 did not induce bleeding even at the highest concentrations tested possibly due to the fact that it does not change the function of coagulation proteins, unlike conventional cationic polymers or cationic proteins[33,64–66]. The low charge density of MPI 8 at physiological pH due to the smart protonation behavior of attached CBG II and the placement of CBGs within the engineered mPEG brush on the polymer scaffold that acts to limit nonspecific binding increases its biocompatibility and selectivity[33,34]. The more predictable bleeding time for MPI 8, which was similar to the saline control, further demonstrates its advantages. Finally, MPI 8 showed remarkable safety based on both acute and chronic toxicity studies in mice. Even at the very high dose of 500 mg/kg, MPI 8 did not induce any changes in liver enzyme levels or organ structures evaluated histologically, suggesting that MPI 8 is very well tolerated. In comparison, polycationic protamine sulfate is lethal at 30 mg/kg in mice[31]. PAMAM and polypropylenimine (PPI) from generations G3 to G5 were shown to be highly cytotoxic[67]. Overall our study validated a concept in the design of a safer antithrombotic.

The lead MPI 8 presents a number of advantages over other polyP inhibitors. As mentioned, conventional polycations (e.g., PEI and PAMAM) are cytotoxic and hemotoxic[64,68]. Peptide based approaches have also been explored and included the use of poly-L-lysine, which can alter fibrin fibril thickness, resulting in an increased risk of thrombosis[69]. Enzyme based approaches such polyphosphatase-based, e.g., PPX[26], can potentially remove phosphates from other phosphate-containing vital molecules in the body. MPI 8 mitigates these adverse interactions and presents high hemocompatibility, and a high dose tolerance. Further, because MPI 8 inhibits polyP through an electrostatic neutralization without degrading polyP, it is likely that MPI 8 will not interfere with critical small-molecule, phosphate containing compounds in vivo. Thus, compared to previous approaches[25–27], MPI 8 demonstrates great potential as a therapeutic, by providing nontoxic thrombo-protection without the bleeding risk. The data presented support the hypothesis that minimized charge density paired with specifically smart binding groups result in targeted polyP inhibition with minimal interference with hemostasis, a significant improvement over current polyP inhibitors. These findings lay the groundwork for a safer and effective antithrombotic.

## Methods

### Ethical statement

All in vitro and in vivo experiments carried out at the University of British Columbia (UBC) in have been approved by the Institutional Review Board (IRB) and UBC animal care committee (ACC). The protocol for blood collection from human subjects in the Centre for Blood Research at UBC has been approved by the Institutional Review Board (IRB) within the University of British Columbia (UBC Ethics approval no: H10-01896) with written consent obtained from donors. Animal experiments involving mouse tail bleeding and thrombosis models carried out at the University of Michigan were performed in accordance with guidelines and were approved by the University of Michigan Care Committee.

### Blood collection and plasma preparation

Blood was drawn from consenting informed healthy volunteer donors at the Centre for Blood Research, University of British Columbia, at 1:9 into BD Vacutainer® Citrate Tubes containing 3.2% buffered sodium citrate solution. Blood was centrifuged at $150 \times g$ for 10 min to separate platelet-rich plasma (PRP), and then spun at $1000 \times g$ for 15 min for platelet-poor plasma (PPP). Pooled normal plasma (PNP) from 20 donors was purchased from Affinity Biologicals (ON, Canada).

### MPI synthesis, modification, conjugation, and characterization

The details of the synthesis of MPI compounds and their characterization (nuclear magnetic resonance spectroscopy (NMR) analysis, molecular weight determination, conductometric measurements, potentiometric measurements, surface plasmon resonance measurements and isothermal titration calorimetry measurements) are given in supplementary methods.

### In vitro analysis of select MPI candidates using microplate assays

**Thrombin generation assay by calibrated automated thrombography for the evaluation of MPI inhibition activity.** A thrombin generation assay (TGA) was carried out at 37 °C by measuring the fluorescence intensity upon cleavage of the fluorogenic thrombin substrate, Z-Gly-Gly-Arg-AMC. Commercially available pooled normal platelet poor plasma (PNP, 30 donors) from George King Bio-Medical, USA was mixed 1:1 with HBS (20 mM HEPES with 100 mM NaCl at pH 7.4). Phosphatidylcholine (80): phosphatidylserine (20) (PCPS) liposomes were added to obtain a final concentration of 20 μM. Serial dilutions of MPI candidates and UHRA were prepared fresh for these experiments. Experiments were repeated twice with two technical replicates each.

Thrombin calibrator was added to each well following the manufacturer's instructions and the thrombin generation assay was initiated by the addition of fluorogenic substrate (both from Diagnostica Stago). Substrate hydrolysis was monitored on a Thrombinograph™ plate reader from Diagnostica Stago. The fluorescence intensity was recorded at 37 °C every 30 s over a period of 1.5 h and analyzed using Thrombinoscope™ software from Diagnostica Stago.

**Determination of the effect of MPI on thrombin generation in a TF-triggered plasma system.** To determine the effect of each MPI candidate (Table 1, main text) on thrombin generation in the absence of polyP, FluCa reagent and TF reagent "PPP" (Thrombinoscope™) at a final concentration of 5 pM were added to initiate thrombin generation. Final concentrations of MPIs ranged from 5–50 μg/mL. No other clotting initiators or accelerators were added.

**Determination of long chain polyphosphate (LC polyP) inhibition by thrombin generation assay.** To validate the effect of LC polyP inhibition, a mixture of plasma, PCPS and MPI or UHRA were incubated with LC polyP (200 μM) for 3 min at 37 °C prior to initiate thrombin generation Concentrations of inhibitors tested ranged from 0.2–100 μg/mL.

**Determination of short chain polyphosphate (SC polyP) inhibition by thrombin generation assay.** To validate the effect of SC polyP inhibition, FXII deficient plasma (Haemtech) was used. To a mixture of plasma, PCPS and MPI, SC polyP was added at a final concentration of 5 μM. TF (Thrombinoscope™) at a final concentration of 8.3 fM was also included in the mixture. The inhibitor concentrations tested ranged from 0.2 to 100 μg/mL.

**Effect on TF-initiated plasma clotting in FXII deficient plasma.** For the clotting assay, each well was filled with 100 μL of a mixture containing FXII deficient plasma (Haemtech) (50 μL), MPI (100 μg/mL, final) in HEPES buffered saline with bovine serum albumin (HBSA, 20 mM HEPES and 100 mM NaCl at pH 7.4 with 0.1% BSA) and relipidated TF in 30% PCPS liposomes. This mixture was incubated for 120 s at 37 °C, then clotting was initiated by addition of 50 μL pre-warmed (37 °C) 25 mM CaCl₂. Clot time was measured on a STart 4® coagulometer (Diagnostica Stago, France).

**Determination of polyP inhibition activity by viscosity-based plasma clotting assay.** For this assay, we followed a recent report from our group[63]. Detailed procedure is given in the supplementary methods.

**Influence of MPI on whole blood clotting by thromboelastometry.** Whole blood was mixed with MPI (MPI 1, MPI 6, and MPI 8 with a final concentration of 100 μg/mL) in HEPES buffered saline (HBS) (20 mM HEPES + 150 mM NaCl) and analyzed for whole blood clotting using a ROTEM® *delta* from Tem Innovations GmbH (Instrumentation Laboratory as of 2016) at 37 °C. Stock solutions of the MPIs and UHRA were prepared at concentrations 100X the final desired concentration in HBS. Citrate anticoagulated whole blood (356 μL) was mixed with 44 μL of the MPIs or UHRA. Three hundred and forty microliters of this suspension were transferred into the ROTEM cup and was re-calcified with 20 μL of 0.2 M calcium chloride solution. HBS mixed with whole blood was used as a negative control for the experiment.

**Influence of fibrin clot structure and fiber diameter by scanning electron microscopy (SEM).** The effects of MPI 8 on fibrin clot structure and fibrin diameter in the presence of MPI 8 were assessed by SEM by following a protocol described previously[33]. Detailed procedure is given in supplementary methods.

### Studies of MPI effects in mice

**Influence of MPI on bleeding in mice without added polyP.** A mouse model was used to assess the effect of MPI on bleeding in the absence of added polyP or any anticoagulants. Heparin was used as a positive control and saline was used as a negative control. Eight-to-ten-week-old C57/BL6 mice were obtained from The Jackson Laboratories (Bar Harbor, ME), and the experimental protocol was approved by the International Animal Care and Use Committee at the University of Michigan (PRO00010481). Light cycles in the animal holding rooms are set for 12 h on and 12 h off. Temperature, humidity and airflow are maintained and controlled. Mice are caged in autoclaved ventilated caging at a capacity of 4 animals/cage during the course of the experiment. Mice were anesthetized, weighed and placed on a heated surgical tray. The tail was immersed into 15 mL of pre-warmed (37 °C) sterile saline (0.9% NaCl). To test the bleeding effects of different agents, MPIs, UHRA-10, saline or heparin were injected retro-orbitally and allowed to circulate for 5 min using solutions of MPI 1, MPI 6, MPI 8, UHRA-10 and UFH in sterile saline for maximum injection volumes of 50 μL and final concentrations of 100-300 mg/kg, or 200 U/kg for UFH. The distal tail (5 mm from the tip) was amputated with a surgical blade (Integra Miltex) and immediately re-immersed in 15 mL of pre-warmed (37 °C) sterile saline (0.9% NaCl). The time required for spontaneous bleeding to cease was recorded. After a maximum of 10 min, the tail was removed from the saline and the mouse was euthanatized by cervical dislocation. The blood samples were then pelleted at 500 × g for 10 min at room temperature and the pellet was resuspended in 5 mL of Drabkin's Reagent (Sigma) and incubated at room temperature for 15 min. The amount of hemoglobin lost was quantified by comparing the absorbance of the samples at 540 nm to a standard curve of bovine hemoglobin in Drabkin's reagent.

**Laser ablation saphenous vein hemostasis model.** Adult wild type mice (male ten- to twelve-week-old C57/BL6 mice were obtained from The Jackson Laboratories (Bar Harbor, ME)) were intravenously injected with 200 mg/kg MPI 8 or equivalent volume of PBS via tail vein injection. Light cycles in the animal holding rooms are set for 12 h on and 12 h off. Temperature, humidity and airflow are maintained and controlled. Mice are caged in autoclaved ventilated caging at a capacity of 4 animals/cage during the course of the

experiment. The effect of the MPI on hemostatic clot formation in vivo was examined using laser ablation saphenous vein hemostasis model as described[48,49]. Briefly, mice were anesthetized by an intraperitoneal injection of ketamine/xylazine (100 and 10 mg/kg, respectively) and intravenously administered DyLight 488-conjugated rat anti-mouse platelet GP1bβ antibody (0.1 μg/g; EMFRET Analytics) and Alexa Fluor 647-conjugated anti-fibrin (0.3 μg/g) via tail vein cannula. The saphenous vein was surgically prepared under a dissecting microscope and superfused with preheated bicarbonate saline buffer throughout the experiment. Blood flow of the saphenous vein was visualized under a 20X water immersion objective using a Zeiss Axio Examiner Z1 fluorescent microscope equipped with a solid laser launch system. The saphenous vascular wall was exposed to two maximum-strength 532-nm laser pulses (70 lJ; 100 Hz; for about 7 ns, 10 ms intervals) to puncture a hole (48–65 μm in diameter) in the vessel wall, resulting in bleeding visualized by the escape of fluorescent platelets to the extravascular space. The laser injury was performed at 30 s and repeated 5 and 10-min after the initial injury at the same site to assess the platelet-fibrin hemostatic clot formation. The dynamics of platelet accumulation and fibrin deposition within the clot were recorded in real-time and the changes in the mean fluorescent intensity over time were analyzed using the Slidebook 6.0 program. A total of 4 mice (2 males and 2 females) with 3–4 independent injuries each were analyzed in the control and treatment groups. DyLight 488-conjugated rat anti-mouse platelet GP1bβ antibody was purchased from EMFRET Analytics. Anti-mouse fibrin antibody for in vivo studies was provided by Dr. Rodney M. Camire at Children's Hospital of Philadelphia and Alexa Fluor 647−conjugated using Alexa Fluor™ Antibody Labeling Kits purchased from Invitrogen based on the manufacturer's instruction.

**Inhibition of thrombosis by MPI in intravital microscopy laser-induced cremaster arteriole thrombosis model.** The effect of the MPIs on thrombosis was examined in mice in real-time under intravital microscopy. Ten to twelve-week-old C57/BL6 mice were obtained from The Jackson Laboratories (Bar Harbor, ME). Light cycles in the animal holding rooms are set for 12 h on and 12 h off. Temperature, humidity and airflow are maintained and controlled. Mice are caged in autoclaved ventilated caging at a capacity of 4 animals/cage during the course of the experiment. The experimental protocol was approved by the International Animal Care and Use Committee at the University of Michigan. The dynamic accumulation of platelets and fibrin within thrombi at the site of the injury in vivo was evaluated in cremaster arterioles in response to laser injury under intravital microscopy. Briefly, male adult mice were anesthetized, the jugular vein was cannulated, a tracheal tube was inserted to facilitate breathing, and the cremaster muscle was surgically prepared and perfused with preheated bicarbonate-buffered saline throughout the experiment under intravital microscopy. DyLight 488- conjugated rat anti-mouse platelet GP1bβ antibody (0.1 μg/g; EMFRET Analytics) and Alexa Fluor 647- conjugated anti-fibrin (0.3 μg/g) were administered by a jugular vein cannula to fluorescently label circulating platelets and detection of fibrin in vivo. 100 mg/kg of MPI 1, MPI 6, and MPI 8 or control buffer were intravenous injected into mice 10 min prior to vascular injury. Multiple independent thrombi (average of 8 thrombi in each mouse, 3 mice per group) were induced in the arterioles (30–50 μm diameter) in each mouse by a laser ablation system (Ablate! photoablation system; Intelligent Imaging Innovations, Denver, CO, USA). Images of thrombus formation at the site of injured arterioles were acquired in real time under 63× water-immersion objective with a Zeiss Axio Examiner Z1 fluorescent microscope equipped with solid laser launch system (LaserStack; Intelligent Imaging Innovations) and high-speed sCMOS camera. All captured images were analyzed

for the change of fluorescent intensity over the course of thrombus formation after subtracting fluorescent background using the Slidebook program.

**Inhibition of thrombosis by MPI via FeCl₃-induced injury to carotid arteries.** MPI and UHRA compounds were evaluated in FeCl₃-induced injury to carotid arteries in mice using a previously reported protocol[32]. The compounds were injected retro-orbitally. Detailed protocol is given in supplementary methods.

**Inferior vena cava (IVC) stenosis (partial) ligation model of thrombosis.** Twenty male C57BL/6 mice were used to investigate the effect of MPI 8 on venous thrombogenesis. Light cycles in the animal holding rooms are set for 12 h on and 12 h off. Temperature, humidity, and airflow are maintained and controlled. Mice are caged in autoclaved ventilated caging at a capacity of 4 animals/cage during the course of the experiment. Alzet 1003D micro osmotic pumps (Cupertino, CA) were placed in a subcutaneous location to deliver MPI 8 ($n = 10$) or control vehicle ($n = 10$) at a rate of 100 μg/h -18-24 h prior to venous thrombosis (VT) induction. The St. Thomas Stenosis Model to induce VT was performed using a 30 g needle spacer and ligation of side branches, but not back branches, as described[46,47] (Reference: PMID 30786739 and PMID 31887775). The dosage times vary because the installation time of the Alzet pump is much less than the stenosis surgery time. Harvest of the thrombosed IVC segment for analysis was at 48 h.

## Tolerance of MPI 8

**Acute toxicity in mice.** The study was performed at the University of British Columbia. Animal toxicity studies were approved by the Institutional Review Board with UBC Ethics approval number A18-0276. An escalating dose study in mice was used to measure the tolerance of MPI 8. Female Balb/C mice (6–8 weeks, 20–26 g) were individually weighed and were divided into groups of 4 for each dose. Light cycles in the animal holding rooms are set for 12 h on and 12 h off. Temperature, humidity and airflow are maintained and controlled. Mice are caged in autoclaved ventilated caging at a capacity of 4 animals/cage during the course of the experiment. Each group of mice ($N = 4$) were treated intravenously (via tail vein) with increasing doses of MPI 8 (250–500 mg/kg). The injection volume was 200 μL/20 g mouse. The mice were briefly restrained (<30 s) during i.v. injections. Dilation of the vein was achieved by holding the animals under a heat lamp for about 1–2 min. After injection, the mice were returned to the cages and monitored for signs of acute toxicity over a period of 1 day. Body weights of individual mice were recorded prior to injection. After 24 h of injection, mice were terminated by $CO_2$ asphyxiation, blood (50 μL) was collected from each mouse on the final day and necropsy was performed on all animals. Serum samples were analyzed for lactate dehydrogenase (LDH), aspartate aminotransferase (AST) and alanine aminotransferase (ALT) activity.

**Analysis of LDH activity in mouse serum samples.** Serum samples were analyzed for LDH activity using a lactate dehydrogenase enzyme assay kit (Abcam., Cambridge, UK). The kit measures the concentration of LDH using a direct, plate-based, colorimetric titration and consists of a 96-well microtiter plate, LDH reagent mix, standard and standard dilution buffer. When serum is added to the LDH reagent mix, the LDH in the sample converts the lactate and NAD+ in the mix to pyruvate and NADH, which interacts with a specific probe to produce a color that can be monitored by measuring the increase in the absorbance of the reaction at 450 nm over a 5 min time interval. In a typical test procedure, 5 μL of the serum sample (dilution factor determined upon initial reading) was added

in duplicate to microplate wells and incubated with 50 μL of the reconstituted LDH (as per the supplier's instructions) and the absorbance was measured at 450 nm. The quantitative assay kit has a detection limit of >1 mU/mL and a range of 1–100 mU/mL. A calibration curve was created using standards of NADH from 0 to 12.5 nmol/well. The average value of the absorbance was used in combination with the standard curve to obtain the LDH activity (U/mL).

**Analysis of AST activity in mouse serum samples.** Serum samples were analyzed for AST activity using an aspartate aminotransferase enzyme assay kit (Sigma Aldrich., Oakville, ON, catalogue # MAK055). The kit measures the concentration of AST using a direct, plate-based, colorimetric titration and consists of a 96-well microtiter plate, AST reagent mix, standard and standard dilution buffer. When serum is added to the AST reagent mix, the AST in the sample transfers an amino group from aspartate to α-ketoglutarate resulting in oxaloacetate and glutamate, which results in the production of a colorimetric product proportional to the AST enzymatic activity present. This activity can be monitored by measuring the increase in the absorbance of the reaction at 450 nm over a 30 min time interval. In a typical test procedure, 50 μL of the serum sample (dilution factor determined upon initial reading) was added in duplicate to microplate wells and incubated with 100 μL of the reconstituted AST reaction mixture (as per the supplier's instructions) and the absorbance was measured at 450 nm. A calibration curve was created using standards of glutamate from 0 to 10 nmol/well. The average value of the absorbance was used in combination with the standard curve to obtain the AST activity (IU/mL).

**Analysis of ALT activity in mouse serum samples.** Serum samples were analyzed for ALT activity using an alanine aminotransferase enzyme assay kit (Sigma Aldrich., Oakville, ON, catalogue # MAK052). The kit measures the concentration of ALT using a direct, plate-based, colorimetric titration and consists of a 96-well microtiter plate, ALT reagent mix, standard and standard dilution buffer. When serum is added to the ALT reagent mix, the ALT in the sample transfers an amino group from alanine to α-ketoglutarate resulting in pyruvate and glutamate, which results in the production of a colorimetric product proportional to the ALT enzymatic activity present. This activity can be monitored by measuring the increase in the absorbance of the reaction at 570 nm over a 30 min time interval. In a typical test procedure, 20 μL of the serum sample (dilution factor determined upon initial reading) was added in duplicate to microplate wells and incubated with 100 μL of the reconstituted ALT reaction mixture (as per the supplier's instructions) and the absorbance was measured at 570 nm. A calibration curve was created using standards of pyruvate from 0 to 10 nmol/well. The average value of the absorbance was used in combination with the standard curve to obtain the ALT activity (IU/mL).

**Chronic toxicity of MPI 8.** Animal toxicity studies were approved by the Institutional Review Board with UBC Ethics approval number A18-0276. An escalating dose study in mice was used. Female Balb/C mice (6–8 weeks, 20–26 g) were individually weighed and were divided into groups of 4 for each dose. Light cycles in the animal holding rooms are set for 12 h on and 12 h off. Temperature, humidity and airflow are maintained and controlled. Mice are caged in autoclaved ventilated caging at a capacity of 4 animals/cage during the course of the experiment. Cages are changed bi-weekly. Each group of mice ($N = 4$) were treated intravenously (via tail vein) with increasing doses of MPI 8 (100–500 mg/kg). The injection volume was 200 μL/20 g mouse. The mice were briefly restrained (<30 s) during i.v. injections. Dilation of the vein was

achieved by holding the animals under a heat lamp for about 1–2 min. After injection, the mice were returned to cages and monitored daily for signs of toxicity over a period of 14 days. Body weights of individual mice were recorded prior to injection and every day except weekends thereafter. On day 15, mice were terminated by $CO_2$ asphyxiation, blood (50 μL) was collected from each mouse on the final day and necropsy was performed on all animals. Serum samples were analyzed for lactate dehydrogenase (LDH) activity using a lactate dehydrogenase enzyme assay kit (Abcam., Cambridge, UK).

## Statistics and reproducibility

Data are expressed as the mean ± standard deviation from $n$ (≥3) independent experiments unless otherwise specified. Statistical analyses were performed using GraphPad Prism version 7.0 software, using a Student's $t$-test or by one-way ANOVA followed by a Dunnett post hoc test which is specified under specific experimental protocols. $p$ values <0.05 were considered statistically significant. Samples were denoted as statistically significant. $*p < 0.05$, $**p < 0.01$, $***p < 0.001$, and $****p < 0.0001$. Experiments were performed in duplicate at least three times and results were pooled into a single dataset unless stated otherwise.

## Reporting summary

Further information on research design is available in the Nature Portfolio Reporting Summary linked to this article.

## Data availability

The data that support the findings of this study is available within the main text and its Supplementary Information file. Data is also available from the corresponding author upon request.

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

## Acknowledgements

We thank the Macromolecular Hub, CBR for the use of their research facilities and access to the UBC bioimaging facility. We acknowledge funding from the Canadian Institutes of Health Research (CIHR) to J.N.K. E.M.C., C.A.H., and the Natural Sciences and Engineering Council of Canada (NSERC; to J.N.K.). We acknowledge funding from National Institutes of Health grants R35 HL135823 and UM1 HL120877 (to J.H.M.), R01 HL144550 (to P.K.H.), and R35 GM131835 (to M.H.). The infrastructure facility is supported by the Canada Foundation for Innovation (CFI) and the British Columbia Knowledge Development Fund (BCKDF). J.N.K. held a Career Investigator Scholar award from the Michael Smith Foundation for Health Research (MSFHR). J.N.K. is a Tier 1 Canada Research Chair in Immunomodulating Materials and Immunotherapy. C.C.L acknowledges the NSERC CREATE NanoMat Program and H.D.L. acknowledges funding from NSERC CGS-M and the NSERC CREATE NanoMat Program. We thank Nancy Dos Santos of BC Cancer Agency, Vancouver for performing some animal studies.

## Author contributions

C.C.L., S.A.S., C.A.H., J.H.M., and J.N.K. designed the experiments and wrote the manuscript. C.C.L. performed synthesis, polyP inhibition, blood compatibility, animal studies and analyzed the data. S.A.S. performed thrombin generation and part of the animal experiments. S.V. contributed to the experimental design and performed experiments. R.A. and N.R. performed intravital microscopy analysis and saphenous vein bleeding analysis supervised by M.H. C.E.L. performed IVC model studies supervised by P.K.H. S.A., E.M.C., S.K.S., M.G.J., M.T.K., and L.A.C. contributed to the experimental design and analyzed the data. I.C. contributed to the synthesis, H.D.L. performed clot structure analysis and M.D. performed ADP binding studies. C.D. performed histology analysis. All authors contributed to editing the manuscript. J.H.M. provided supervision and grant support. J.N.K. conceived the project, provided supervision and grant support for the project.

## Competing interests

The University of British Columbia and the University of Michigan have filed a patent application on the work described here, with C.C.L., S.A.S., S.V., C.A.H., J.H.M., and J.N.K. as co-inventors. International Application No.: PCT/US2022/044259 Filed: 21-Sep-2022. The remaining authors declare no other competing interests.

## Additional information

[1]Centre for Blood Research, Life Sciences Institute, University of British Columbia, Vancouver, BC, Canada. [2]Department of Chemistry, University of British Columbia, Vancouver, BC, Canada. [3]Department of Biological Chemistry, University of Michigan Medical School, Ann Arbor, MI, USA. [4]Department of Pathology and Laboratory Medicine, University of British Columbia, Vancouver, BC, Canada. [5]Department of Pharmacology, University of Michigan Medical School, Ann Arbor, MI, USA. [6]Department of Surgery, Section of Vascular Surgery, University of Michigan Medical School, Ann Arbor, MI, USA. [7]Department of Chemical and Biological Engineering, University of British Columbia, Vancouver, BC, Canada. [8]Michael Smith Laboratories, University of British Columbia, Vancouver, BC, Canada. [9]Bloodworks Research Institute, 1551 Eastlake Avenue E.; Ste.100, Seattle, WA 98102, USA. [10]Department of Urological Sciences, University of British Columbia, Vancouver, BC, Canada. [11]Department of Medicine, University of British Columbia, Vancouver, BC, Canada. [12]The School of Biomedical Engineering, University of British Columbia, Vancouver, BC, Canada. [13]Present address: Bloodworks Research Institute, 1551 Eastlake Avenue E.; Ste. 100, Seattle, WA 98102, USA. ✉e-mail: jhmorris@umich.edu; jay@pathology.ubc.ca

