## [Peer Review File · Nature Communications]

Smart Thrombosis Inhibitors Without Bleeding Side Effects via Charge Tunable Ligand DesignEditorial Note: In this file, Figure 2 (A1, B1, C1), Figure 3 (A1), Figure 4 (B1), Figure 5 (A1) were created by BioRender.com

REVIEWER COMMENTS

Reviewer #1 (Remarks to the Author):

This manuscript reports on the use of different cationic synthetic polymeric nanostructures as binders of inorganic polyphosphate (polyP), and inhibitors of the effect of the polyP on the blood coagulation. The blood coagulation is a complex biological process that is still poorly understood. The modulation of this process is fundamental for the treatment and control of pathological situations and key for clinical management. Thus, any development in this area is of great general interest. Moreover, the specific effect of polyP in these processes is still unclear and this work further deepens on that topic. Accordingly, my overall impression is that the manuscript merits to be published as an important research problem is faced. Despite the biological part of the work does not overlaps my specific expertise, it seems to be professionally done and the results are in line with the general conclusions. However, I have several concerns about the presentation of the binding data and its novelty.

1) First of all, this part of the manuscript strongly overlaps with the unpublished material included as an additional paper to be submitted elsewhere. This paper (also available to reviewers, despite I had to ask for it) shows important overlapping sections (tables and figures) with the current submission to Nature Communication, which makes me wonder if such a fragmentation is necessary or even positive to understand the current submission. Actually, a key idea argued as the main point of the design of the molecules (change of protonation degree upon polyP binding at neutral pH) is not clearly presented in the current submission but explained in the unpublished additional materials intended to be submitted to a different journal. The overall situation makes the reviewer (and probably the future reader) to feel uncomfortable. The authors should consider merging both papers or, at least, submitting the biological activity results once the binding of the systems has been published and publicly accessible.

2) I also have some concerns about the novelty of the design vectors. The authors claim that the originality of their design lies on the implementation of pKa close to neutrality for some amino groups, which will be protonated upon polyP binding, thus making the systems less charged (and less toxic) in the absence of polyP. This concept is indeed appealing and worth to be mentioned, but from the supramolecular chemistry point of view, this is not new at all. The change in the local protonation state upon anion/cation binding was already observed in the infancy of supramolecular chemistry of polyamines by pioneers such as Vacca, Martell, or even the Nobel Laureate Jean Marie Lehn in the 80's of last decade. Actually this is the key issue for measuring anion/cation binding with polyamines by potentiometric titrations. I would reduce the novelty claims and I would also suggest the authors to give credit to some of the seminal studies.

3) Considering point 2 and that the authors seem to have access to potentiometric titrations as a regular technique, I wonder why they did not use this technique to measure the actual binding, since this is very sensitive to changes in the protonation degree. The most convincing evidence of a hypothesis must use the most sensitive experimental measurement of the involved property.

4) The SPR studies show the traces of titrations using the steady-state approach. However, for some complex systems, this is not accurate if the on/off kinetics are not very fast. The actual SPR traces must be included in the supporting information and, in case the kinetics are not fast enough, the kon/koff kinetic fitting must be used for the binding. This is especially relevant for molecules to be used in biological media, since in such a complex and competitive environment, the corresponding equilibrium state is difficult to achieve. Therefore, values of kon/koff are extremely informative.

In summary, I cannot support acceptance of the manuscript in the current form. My first suggestion would be to merge the two papers in a more complete and general study. Alternatively, the authors could consider keep the fragmentation, but in that case, the paper describing the binding of the systems must be published before for a more suitable description of the process, and the impact and general interest of the corresponding manuscript would be lower. In any case, the additional suggestions might be also strongly considered.

Reviewer #2 (Remarks to the Author):

This is an interesting manuscript describing the next generation of cationic molecules that inhibit the activation of the contact pathway.

Major comments.

- 1/ The manuscript focuses on polyP for the in vitro studies of coagulation. However, it is likely that there are additional negatively-charged molecules that activated the contact pathway in vivo. Therefore, the authors should determine if the MPI candidates also inhibit the activation of the contact pathway by other activators, such as kaolin and silica.
- 2/ Thrombosis was examined using the laser-induced cremaster arteriole thrombosis model and the ferric chloride artery injury model. A better model to examine the antithrombotic effects of MPI 8 is the inferior vena cava model. This is a better model of VTE in patients.
- 3/ MPI 8 did not cause an increase in bleeding in the tail transection model. It would be helpful to examine the effect of MPI 8 in other bleeding models, such as the saphenous vein model.
- 4/ Knock out of the inositol kinase in mice was associated with reduced generation of polyP and increased tail bleeding indicating a role of polyP in hemostasis. Why does an inhibitor of polyP not cause similar bleeding? FXI is also required for hemostasis in humans.
- 5/ Serum levels of LDH were used as a measure of toxicity. Liver enzymes should also be measured. How are MPI-polyP complexes removed from the circulation?
- 6/ What small molecule phosphate containing compounds are present in vivo?

Reviewer #3 (Remarks to the Author):

This is a concise, well compiled, and written manuscript comprising the in-vitro and in-vivo characterization of macromolecular polyanion inhibitors.

The quality of the data is appropriate and technically sound. Presentation and interpretation of the data as well as controls are in large parts of the manuscript sufficient.

The presented data reveal sufficiently strong evidence for the author's claims.

The report is of interest for the community and can be published – according to the opinion of this reviewer – with minor changes as depicted in the following comments.

Comments:

#1 According to the ISO norm - e.g. 10993-4, Biological evaluation of medical devices - Part 4: Selection of tests for interactions with blood - hemocompatibility testing includes analyses of blood platelet and leucocyte as well as complement activation. None of these recommended tests were conducted in this study. Thus, I recommend specifying the terms used to describe the conducted analyses more precisely. The same applies to "Biocompatibility".

The authors state "superior hemocompatibility".

In my very personal opinion, superlatives should be avoided in a scientific manuscript to ensure an appropriate scientific discussion.

#2 Please describe which mechanisms might be responsible for the reduced bleeding time (MPI 8 – 100, MPI 8, 300) in the mouse model. The differences are not significant, however, a tendency is clearly indicated.

#3 Please discuss in the manuscript the increase in the LDH values as well (Fig 5 C3). Despite differences are tendential only, the data show an outlier in the saline control, which might be the reason why differences between the samples are not significant.

#4 Abbreviations such as ALT and LDH are not explained in the main text. Since the Material and Methods section is provide as a Supplement, I recommend including the explanations in the main text

as well.

#5 Please correct the error message in the 1.1.1 Materials and Reagents:

The protocols for synthesis of UHRA 8, UHRA 10 and MPis studied are provided in 1.1.2 and the physical characteristics are shown in Error! Reference source not found.. BD Vacutainer® Citrate Tubes containing 3.2 % buffered sodium citrate solution were purchased from Becton, Dickinson and Company (New Jersey, USA).

#6 Section 1.1.1.3 Blood collection and plasma preparation

The authors state “healthy volunteer donors”. How was the health status of the human donors determined? Which were the inclusion criteria for the donors? Which tests were conducted to exclude inflammatory processes and appropriate hemostatic/cell function? Were the donor data compared with reference values for apparently healthy humans? The number of donors included in the study (not pooled plasma) should be stated in the materials and methods section.

Some of these data are already in the Reporting Summary. I recommend stating this in the manuscript as well.

Characterization of the donors should include the collection of demographic data, data about the recent intake of medication, special nutrition and extensive physical activity. Participants should be excluded from the study if any of these is reported to affect the hemostatic system such as platelet function inhibitors, diabetic and allergic disorders and respective medication as well as excessive consumption of certain food or extensive sports. For thrombogenicity and hemocompatibility studies, characterization of the donated blood should comprise e.g. whole blood haemogram data, analysis of markers of acute inflammation, e.g., C-reactive protein levels as well as test for platelet function for achieving reproducible test results.

Response to reviewer's comments:

REVIEWER COMMENTS

Reviewer #1 (Remarks to the Author):

Comment: This manuscript reports on the use of different cationic synthetic polymeric nanostructures as binders of inorganic polyphosphate (polyP), and inhibitors of the effect of the polyP on the blood coagulation. The blood coagulation is a complex biological process that is still poorly understood. The modulation of this process is fundamental for the treatment and control of pathological situations and key for clinical management. Thus, any development in this area is of great general interest. Moreover, the specific effect of polyP in these processes is still unclear and this work further deepens on that topic. Accordingly, my overall impression is that the manuscript merits to be published as an important research problem is faced. Despite the biological part of the work does not overlaps my specific expertise, it seems to be professionally done and the results are in line with the general conclusions. However, I have several concerns about the presentation of the binding data and its novelty.

Response: We thank the reviewer for carefully going through the manuscript and positive outlook on the data presented. We have revised our manuscript to fully address all the comments raised.

Comment #1. First of all, this part of the manuscript strongly overlaps with the unpublished material included as an additional paper to be submitted elsewhere. This paper (also available to reviewers, despite I had to ask for it) shows important overlapping sections (tables and figures) with the current submission to Nature Communication, which makes me wonder if such a fragmentation is necessary or even positive to understand the current submission. Actually, a key idea argued as the main point of the design of the molecules (change of protonation degree upon polyP binding at neutral pH) is not clearly presented in the current submission but explained in the unpublished additional materials intended to be submitted to a different journal. The overall situation makes the reviewer (and probably the

future reader) to feel uncomfortable. The authors should consider merging both papers or, at least, submitting the biological activity results once the binding of the systems has been published and publicly accessible.

Response: We made substantial revision to address this comment. In fact, we overlooked the overlap after completing the editing of the second manuscript. Thus, we have combined the two manuscripts to one. We have enhanced design concept, added information about pKa of different ligands, potentiometric titration data, demonstration of switch-charge state in revised version of the manuscript. We believe that now it is a full story detailing the development of a new class of molecules, physical concepts around the ligand design and scaffold design, experimental studies detailing the physical mechanism and the biological evaluation of the molecules. Please see pages 4-7, figure 2, supplementary information, Table S2.

Figure 1. Observed enthalpy in different buffer environments compared to heat of ionization shows recruitment of protons upon MPI binding to polyP. a) Evaluation of free unbound MPI charge content. Potentiometric titrations were used to evaluate the charge content of the MPI library at 25 °C and 160 mM NaCl. A sample titration of 0.15 M NaOH into a solution of acidified MPI 3 while measuring the change in potential is shown. b) The

speciation plot calculated from titration of MPI 3 with NaOH. **c)** $\Delta H_{\text{observed}}$ obtained from binding MPI 3 and polyP (P75) using ITC200 in four buffers plotted against each buffers' $\Delta H_{\text{ionization}}$, at pH 7.4, 150 mM NaCl and 25 °C.

Table S1. Summary of log K values obtained from potentiometry for CBGs loaded on MPIs, before and after conjugation.

	$L + H \rightleftharpoons HL$	$HL + H \rightleftharpoons H_2L$	$H_2L + H \rightleftharpoons H_3L$	$H_3L + H \rightleftharpoons H_4L$
CBG I	9.22 ± 0.02	8.41 ± 0.02	2.09 ± 0.02	-
MPI 2	8.7 ± 0.4	7.5 ± 0.2	4.3 ± 0.1	-
CBG II	10.6 ± 0.2	8.3 ± 0.2	4.1 ± 0.1	-
MPI 4	8.9 ± 0.2	6.5 ± 0.3	3.6 ± 0.1	-
Me₆-TREN	9.50	8.68	7.44	< 1.0
UHRA-8	9.2 ± 0.2	7.5 ± 0.1	4.1 ± 0.1	< 1.0

Potential in mV was measured as standardized base (0.15 M NaOH) and titrated into acidified (pH 2) solution of CBG or MPI at 25 °C, 160 mM NaCl. Relatively similar values were obtained for other MPIs.

Comment #2: I also have some concerns about the novelty of the design vectors. The authors claim that the originality of their design lies on the implementation of pKa close to neutrality for some amino groups, which will be protonated upon polyP binding, thus making the systems less charged (and less toxic) in the absence of polyP. This concept is indeed appealing and worth to be mentioned, but from the supramolecular chemistry point of view, this is not new at all. The change in the local protonation state upon anion/cation binding was already observed in the infancy of supramolecular chemistry of polyamines by pioneers such as Vacca, Martell, or even the Nobel Laureate Jean Marie Lehn in the 80's of last decade. Actually, this is the key issue for measuring anion/cation binding with polyamines by potentiometric titrations. I would reduce the novelty claims and I would also suggest the authors to give credit to some of the seminal studies.

Response: Thank you very much for this important comment. We have revised the manuscript thoroughly. In no case, do we want to discount the seminal work by pioneers in the field of polyamines and polyamine biology and supramolecular chemistry. In fact, that previously published work laid the foundation for this work (some of them were cited previously) which has been well cited now in the revised manuscript. Although there are tons of research available on polycations, its binding to polyanion species, and utilization of

such concepts in the design of molecules that target tumors, delivery of nucleic acids therapy etc., polycation toxicity still is a major concern. Previous polycationic systems utilized an external trigger (chemical or physical) in order to fulfill their target binding. No polycation design is currently available that itself act as a therapeutic, and the polycation changes its charge state upon binding to its target itself without the need of an external trigger -- the charge switching concept. Such design concept towards a therapeutic has not been reported previously. In addition to the charge switching concept, the unique features of the MPI's scaffold serve to improve biocompatibility, presumably (at least in part) by limiting nonspecific interactions through the steric repulsion created by the PEG corona (brush layer) on the surface of the MPI. Importantly, the design platform and underlying synthesis chemistry are flexible, potentially enabling specific tuning of the MPI's properties and clinical efficacy through, for instance, the facile synthesis of a library of compounds of varying scaffold size, CBG structure, and quantity and density of charge. Most importantly, utilizing these physical principles, we designed a novel therapeutic with superior safety profile (a most biocompatible polycation) with target specificity. All these add to the novelty of the current work. Please see the changes in Results section pages 4-7 and discussion section page 11.

Comment #3: Considering point 2 and that the authors seem to have access to potentiometric titrations as a regular technique, I wonder why they did not use this technique to measure the actual binding, since this is very sensitive to changes in the protonation degree. The most convincing evidence of a hypothesis must use the most sensitive experimental measurement of the involved property.

Response: We thank the reviewer to bringing this point. As the reviewer pointed out, the technique is not a very regular technique used in our laboratory. However, we have considered and made careful analysis on the use of potentiometric titrations to determine the actual binding of polyP to MPI.

The literature describing potentiometric titrations to determine the binding constant to a macromolecule to small molecule is very well developed both technique wise as well as theoretically. However, the interactions between macromolecule-macromolecule are not well developed based on our best of the current knowledge (Ramamurthy N, et al *Analytical Biochemistry* **266**, 116-124 (1999), Durust N et al *Journal of Electroanalytical Chemistry*, **602**, 138-141 (2007) and Buchanan SAN, et al *Analytical Chemistry* **76**, 1474-1482 (2004)). There are only three papers reported in 1997, 2004 and 2007 on this subject from the best of our knowledge. However, the technique is very well developed to study macromolecule-small molecule interactions and also macromolecule to membrane interactions.

The scientific challenges for us on the use of potentiometric titrations to determine the binding constant are characterized as following.

- The reported literature utilizes a non-equilibrium method to arrive the binding constants (Ramamurthy N, et al *Analytical Biochemistry* **266**, 116-124 (1999), Durust N et al *Journal of Electroanalytical Chemistry*, **602**, 138-141 (2007) and Buchanan SAN, et al *Analytical Chemistry* **76**, 1474-1482 (2004)) which is very sensitive to time, concentration of the reagents etc. This would in fact affect the actual binding constant measured.

- The measurements need a very special type of polyion sensitive membrane electrodes. Binding constant measurements are also sensitive to the type of electrodes. Electrodes that are sensitive to MPIs and polyP have not been developed or tested yet. The response of these electrodes is attributed to a nonequilibrium steady-state change in the phase boundary potential at the membrane/sample interface owing to a cooperative extraction of the polyion in the polymeric membrane phase. It is not clear whether the two membrane electrodes reported, tridodecylmethylammonium chloride (TDMAC) and dinonylnaphthalene sulfonate (DNNS) are sensitive to polyP and MPIs (Ferguson S SA, et al *Sensors and Actuators B: Chemical*, 272, 643-654 (2018); Ramamurthy N, et al *Analytical Biochemistry* **266**, 116-124 (1999), Durust N et al *Journal of Electroanalytical Chemistry*, **602**, 138-141 (2007) and Buchanan SAN, et al *Analytical Chemistry* **76**, 1474-1482 (2004)). In published literature, researchers used strong unshielded polycations such as protamine, polybrene, polyL-arginine, poly(amidoamine), poly(propylenimine) (highly positively charged molecules), but our MPI is a charge protected weakly basic polycation. This would make it challenging to find suitable membrane electrode which is very sensitive to MPI. Thus, there is lot of uncertainty in embarking potentiometric method to determine the actual binding constant.
- In addition, the reported ion-selective membrane electrodes are only sensitive to low concentrations of polyions and has a narrow concentration range sensitivity.

Thus, we have decided against utilizing potentiometric titrations to determine the binding constant due to scientific reasons such as the measurements are not done at true equilibrium and electrode chemistry is not well developed for the polycation such as MPIs. Conventional potentiometric titrations, such as those described in our work, utilized a glass electrode to measure the pH changes.

In addition to this, we have consulted the use of potentiometric titrations to determine the binding constant of MPI-polyP complex formation with one of the world experts (Dr. Meyerhoff, University of Michigan). Dr. Meyerhoff published some of the first papers on the use of potentiometric titration to determine macromolecule-macromolecule interaction and binding constants. Dr. Meyerhoff has a similar opinion ours as described above on this topic.

However, in light of this reviewer comment to determine the binding constant, we have performed several additional experiments utilizing isothermal titration calorimetry (ITC) which provide more realistic and accurate affinity constants at equilibrium. The ITC measurements provide the association constant (K_a) along with the free energy (ΔG), the enthalpy (ΔH) and stoichiometry (N) of the binding reaction, giving deeper insight into the driving force of the complexation reaction between MPI-PolyP. ITC technique is a highly sensitive experimental measurement to study the interaction between macromolecules, and is readily applicable to macromolecule-macromolecule (here MPI-polyP) interactions described in the manuscript. Unlike potentiometric method, it is an equilibrium method and values obtained can be trusted with high confidence.

We have updated the manuscript with following details, binding constant for MPI-polyP interaction, ΔH , ΔS and ΔG for the binding reaction between polyP45 with different MPIs. In fact, we have seen that the weak amines such as MPI show similar affinity constants as strong basic amines (UHRA type) which again confirms our concept that low charge density of MPI at physiological pH is sufficient to generate initial binding to polyP, and the

protonation of the MPI during the interaction with polyP increased the stability of the MPI-polyP complex. The manuscript has been modified to include these data in the revised version.

Please see page 6, Table 2, Table 3S and Figure S5 of the revised manuscript.

Table 2. Summary of ITC data characterizing the thermodynamic properties, stoichiometry (N) and equilibrium dissociation constant K_d for binding of MPI to polyP at pH 7.4 and 25 °C.

polyP ^a	MPI ^b	N ^c	K_d (μ M)	ΔG (kcal/mol)	ΔH (kcal/mol)	$T\Delta S$ (kcal/mol)
P45 ^d	MPI 3	0.7	0.737 \pm 0.01	-8.37 \pm 0.04	-107.0 \pm 0.8	-98.7 \pm 0.8
	MPI 5	0.8	0.95 \pm 0.05	-8.22 \pm 0.02	-75.0 \pm 0.4	-66.8 \pm 0.4
	MPI 7	1.0	1.27 \pm 0.09	-8.05 \pm 0.06	-37 \pm 1	-29.0 \pm 0.9
	MPI 9	1.1	1.2 \pm 0.2	-8.08 \pm 0.02	-50.9 \pm 0.5	-42.9 \pm 0.5

^a Used in the ITC cell.

^b Added to cell via syringe.

^c Ratio of MPI to polyP.

^d Buffer used was sodium phosphate buffer composed of dibasic phosphate buffer (Na₂HPO₄), monobasic phosphate buffer (NaH₂PO₄) and NaCl. NaCl concentration is 10 mM.

Table S3. Summary of binding thermodynamics, stoichiometry and binding affinity data obtained by isothermal titration calorimetry (ITC) at pH 7.4 and 25 °C.

Buffer	NaCl (mM)	Cell (titrand)	Syringe (titrant)	N	K_d (μ M)	ΔG (kcal/mol)	ΔH (kcal/mol)	$T\Delta S$ (kcal/mol)
HEPES ^c	10	P45	CBG I	5.1	7194	-2.92	-2.06	0.87
HEPES ^c	10	P45	MPI 9	1.1	0.49 \pm 0.03	-8.66 \pm 0.04	-37.9 \pm 0.9	-29.2 \pm 0.9
HEPES ^c	10	P45	MPI 3	0.7	0.60 \pm 0.09	-8.50 \pm 0.08	-74.8 \pm 0.8	-66.3 \pm 0.4

^c HEPES buffer consisted of HEPES (4-(2-hydroxyethyl)-1-piperazineethanesulfonic acid) and NaCl.

Figure S5: Thermograms and differential binding curves obtained by ITC for select MPI binding to polyP with 10 mM NaCl added, as labelled. All experiments were conducted at pH 7.4 and 25 °C. One representative titration is shown for each system studied, while the thermodynamic data reported were taken as the mean of three independent titrations, each normalized by their respective heats of dilution.

Comment #4: The SPR studies show the traces of titrations using the steady-state approach. However, for some complex systems, this is not accurate if the on/off kinetics are not very fast. The actual SPR traces must be included in the supporting information and, in case the kinetics are not fast enough, the kon/koff kinetic fitting must be used for the binding. This is especially relevant for molecules to be used in biological media, since in such a complex and competitive environment, the corresponding equilibrium state is difficult to achieve. Therefore, values of kon/koff are extremely informative.

Response: We have added representative actual SPR traces were added to the supplementary information. Readers will have access to this in the revised version. Please see supplementary figure S3 and S4.

Figure S1. Summary binding curves for each MPI binding to SC polyP (P110) using surface plasmon resonance (SPR).

Binding affinities were obtained from the steady state affinity for each MPI for each run and then averaged over three runs. All concentrations were run at 25 °C in HBS running buffer with EDTA and P20 surfactant (N=3 runs).

Curves for Short Chain PolyP with MPI-8 with Heterogeneous Ligand Curve Fitting

Affinity Plots for Short Chain PolyP with MPI-8 from Biacore Software

Mean (se) KD: 198 (36) nM

Figure S4: Representative row SPR binding curves for MPI-8 with short chain polyP.

Comment #5: In summary, I cannot support acceptance of the manuscript in the current form. My first suggestion would be to merge the two papers in a more complete and general study. Alternatively, the authors could consider keep the fragmentation, but in that case, the paper describing the binding of the systems must be published before for a more suitable description of the process, and the impact and general interest of the corresponding manuscript would be lower. In any case, the additional suggestions might be also strongly considered.

Response: We once again thank the reviewer for helping us to improve the quality of the manuscript. We have combined the physical chemistry paper and the biological evaluation paper to one manuscript in the revised version. The design concept is enhanced with more added data and discussion also enhanced. We hope the manuscript will be of interest to diverse readers of the journal.

Reviewer #2 (Remarks to the Author):

Comment: This is an interesting manuscript describing the next generation of cationic molecules that inhibit the activation of the contact pathway.

Response: We thank the reviewer for the highly positive feedback and points to improve the quality of the manuscript. We have revised the manuscript to fully address the comments by this reviewer by performing additional experiments as described.

Major comments.

Comment #1: The manuscript focuses on polyP for the in vitro studies of coagulation. However, it is likely that there are additional negatively-charged molecules that activated the contact pathway in vivo. Therefore, the authors should determine if the MPI candidates also inhibit the activation of the contact pathway by other activators, such as kaolin and silica.

Response: We thank the reviewer for bringing this point. To demonstrate the selectivity of the MPI 8 to polyP, we have assessed the contact pathway activation inhibition of different MPI molecules against a nucleic acid (PolyI:C, a known activator of contact pathway). Our studies described in supplementary information, show that MPI 8 does not inhibit the activation of coagulation by PolyI:C, demonstrating its specificity. Since kaolin and silica are non-physiological, we haven't investigated these surfaces. Please see page 12 (discussion section) and supplementary figure S10.

Figure S10: Inhibition of activation of blood coagulation by nucleic acid (surrogate nucleic acid Poly IC) by MPIs. Screening of inhibitors using plasma clotting assay to measure blood coagulation initiation. The data is represented as percent neutralization with 0% being the Poly IC with no inhibitor and 100% being the plasma with no poly IC and no inhibitor. The solid bars indicate 200 µg/mL and striped bar indicate 100 µg/mL of inhibitor concentration. MPI-8 did not inhibit PolyIC initiated blood coagulation.

Comment #2: Thrombosis was examined using the laser-induced cremaster arteriole thrombosis model and the ferric chloride artery injury model. A better model to examine the antithrombotic effects of MPI 8 is the inferior vena cava model. This is a better model of VTE in patients.

Response: In light of this comment, we have evaluated MPI 8 in inferior vena cava (IVC) stenosis (partial) ligation model in mice. Our results are shown in Figure 5C1, Figure 5C2 and supplementary information (Figure S7 and S8), which indicate that MPI 8 is highly effective in preventing venous thrombosis in this model. These data further support the activity of our new molecule to prevent thrombosis. Please see page 10.

Figure 2. MPI 8 exhibits antithrombotic properties in mice. A) MPI 8 reduced fibrin clot formation and platelet accumulation in mouse cremaster arteriole thrombosis model. A1) Schematic representation of the thrombosis model: Platelets and fibrin are tagged with fluorescent antibodies and can be visualized as they accumulate at the site of injury upon laser-induced injury. A2) Representative images of thrombus growth at 0-3 mins in mice which were treated with saline or MPI 8. A3) Median fluorescence intensity representative of fibrin accumulation over time. A4) Median fluorescence intensity representative of platelet accumulation over time. B) MPI 8 delays time to occlusion in carotid artery thrombosis model. B1) Schematic representation of the thrombosis model. Artery patency was monitored by Doppler flow probe. Injury was induced by topical application of FeCl₃ and patency is plotted versus time, comparing the saline control and MPI 8. UHRA-10 was used as a control. B2) At 100 mg/kg, MPI 8 is more effective delaying time to occlusion. A log rank analysis test shows that the curve of MPI 8 is significantly different from the curve of

UHRA-10 (**P<0.0005). **B3)** At 200 mg/kg, MPI 8 and UHRA-10 have reached a similar level of patency, likely a maximum in this model using these inhibitors. All results shown are mean of n = 8 mice. A log rank analysis test indicates the two curves of MPI 8 compared to the saline control are significantly different (**P<0.005). **C) MPI 8 treatment inhibits thrombus formation in inferior vena cava thrombosis model. C1)** a carton representation of the mouse model. **C2)** Thrombus weights of untreated (vehicle control) and treated (MPI 8) mice. After removal of 2 outliers, data were statistically significantly different at p = 0.0003 in an unpaired T-test with Welch's correction.

Table Analyzed	Thrombus wt/ln
Column B	MPI-8 Treatment
vs.	vs.
Column A	Vehicle Control
Unpaired t test with Welch's correction	
P value	0.0201
P value summary	*
Significantly different (P < 0.05)?	Yes
One- or two-tailed P value?	Two-tailed
Welch-corrected t, df	t=2.950, df=7.403
How big is the difference?	
Mean of column A	0.02593
Mean of column B	0.02059
Difference between means (B - A) ± SEM	-0.005338 ± 0.001809
95% confidence interval	-0.009568 to -0.001107
R squared (eta squared)	0.5404
F test to compare variances	
F, DFn, Dfd	34.69, 7, 7
P value	0.0001
P value summary	***
Significantly different (P < 0.05)?	Yes
Data analyzed	
Sample size, column A	8
Sample size, column B	8

Figure S8. Inferior vena cava stenosis thrombosis model. Approximately 18 - 24 hours after MPI 8 was delivered, an IVC stenosis (partial flow) ligation as described (Ref. PMID 30786739 and PMID 26482993) was performed to induce thrombosis. Thrombosis was achieved in 8 of 10 mice for each group (at 48 hours post-VT), and these 8 values were used for comparison. MPI 8 treatment yielded significantly smaller thrombus than control treatment as measured by thrombus weight p=0.0014, and thrombus weight/length p=0.0201. Student' t-Test was used for statistical comparison.

Table Analyzed	Thrombus weight
Column B	MPI-8 Treatment
vs.	vs.
Column A	Vehicle Control
Unpaired t test with Welch's correction	
P value	0.0014
P value summary	**
Significantly different (P < 0.05)?	Yes
One- or two-tailed P value?	Two-tailed
Welch-corrected t, df	t=4.791, df=8.032
How big is the difference?	
Mean of column A	0.02330
Mean of column B	0.01245
Difference between means (B - A) ± SEM	-0.01085 ± 0.002265
95% confidence interval	-0.01807 to -0.005631
R squared (eta squared)	0.7408
F test to compare variances	
F, DFn, Dfd	13.49, 7, 7
P value	0.0028
P value summary	**
Significantly different (P < 0.05)?	Yes
Data analyzed	
Sample size, column A	8
Sample size, column B	8

Figure S9. Inferior vena cava stenosis thrombosis model. **Statistical analysis data for figure 5C2.**

Comment #3: MPI 8 did not cause an increase in bleeding in the tail transection model. It would be helpful to examine the effect of MPI 8 in other bleeding models, such as the saphenous vein model.

Response: We have performed saphenous vein hemostasis model in mice with MPI 8 to demonstrate the safety. Results show that MPI 8 does not increase the bleeding tendency in comparison to the buffer treated animal. These results have been added to Figure 6 B1 and B2. Together with tail vein bleeding model and tolerance studies, support the safety of MPI 8 and the demonstration of the design concept. Please see page 10.

Figure 3. MPI 8 does not induce bleeding in mice. A) High doses of the lead MPI candidates do not cause bleeding in mice. A1) Schematic representation of mouse tail bleeding model with C57/BL6 mice. A2) Recorded bleeding time. Mice were injected in a blinded study with up to 300 mg/kg of lead MPI candidates and demonstrated no increase in bleeding side effect, in contrast to mice administered 200 U/kg of unfractionated heparin. n = 8 mice per group. A3) Hemoglobin lost by mice. B) MPI 8 did not decrease platelet recruitment and fibrin formation in hemostatic clot formation assessed via saphenous vein hemostasis model. Quantitative analysis of the dynamics of platelet accumulation (B1) and fibrin formation (B2) in response to vascular injury in saphenous vein. B3) Representative images of platelet (green) and fibrin (red) hemostatic clot formation in response to a repetitive vascular injury of the saphenous vein. Mice were injected with 200 mg/kg of the lead MPI 8 candidate.

Comment #4: Knock out of the inositol kinase in mice was associated with reduced generation of polyP and increased tail bleeding indicating a role of polyP in hemostasis. Why does an inhibitor of polyP not cause similar bleeding? FXI is also required for hemostasis in humans.

Response: We thank the reviewer for bringing up this point. In the global IP6K1 knockout mouse, the effects are likely not limited to polyP levels in platelets. The potential effects of reduction of polyP production on systems other than hemostasis has not been extensively explored. Furthermore, IP6K1 is a kinase for production of IP7 and IP8 (which are themselves signaling molecules), and it is known that global knockout of IP6K1 in mice has multiple effects on glucose homeostasis, diabetes susceptibility, etc. So, some of the observed effects in this knock-out model could have been unrelated to lower amounts of polyP. Unlike the IP6K knock out, where there is significant reduction in the polyP generated in platelets and elsewhere, we are not decreasing the polyP here in platelets or other cells. Rather, we are neutralizing polyP which is released from the platelets into the blood during the platelet activation that occurs in response to injury. Since polyP is not a *trigger* of hemostatic processes, but rather an *enhancer* of some of the steps of the coagulation cascade, thrombin (and subsequently fibrin) are still generated. Current understanding of the differences between coagulation in hemostasis and that in thrombosis suggest that several of the steps influenced by polyP contribute more to thrombosis than to hemostasis. With regard to FXI, it is true that in humans it participates in hemostasis under some circumstances, but FXI deficiency in mice is not associated with an appreciable bleeding phenotype. On the other hand, pharmacologic targeting of FXI appears to be protective in both animal models of thrombosis and prevention of thrombosis in human patients. We would consequently not expect a marked increase in bleeding when neutralizing polyP.

Comment #5: Serum levels of LDH were used as a measure of toxicity. Liver enzymes should also be measured. How are MPI-polyP complexes removed from the circulation?

Response: Information about liver enzymes has been added in the manuscript (Figure 7 A2, A3). AST and ALT levels are similar to the control animals suggesting there is no liver toxicity

We anticipate that MPI 8 should be excreted predominantly via the kidney as it is below the kidney clearance limit (MPI 8 is ~10 kDa). However, when complexed with polyP, the clearance route might change. We plan to perform a pharmacokinetics study using radiolabeled MPI 8 in the near future and this study is anticipated to answer these questions.

Figure 4. Tolerance of MPI 8 in mice. A) MPI 8 were well tolerated in mice at high doses. A1, A2, A3- LDH activity, ALT activity, AST activity in mice, respectively, after injecting MPI 8 intravenously. Mice were sacrificed after 24 h (acute study) (n=4 mice per group). B1) Schematic representation of mouse chronic toxicity model. Female BALB/c mice in groups of 4 were administered either saline or escalating doses of MPI 8, up to 500 mg/kg. Mice were monitored daily and body weights were measured. After 15 days, serum was collected from sacrificed mice and analyzed for LDH levels. Mice injected with MPI 8 showed no significant change in body weight compared to mice injected with saline and no increase in LDH levels. B2) Change in body weight over 15 days. B3) LDH activity. C) Representative stained (H&E) images of organs collected from mice injected (*i.v.*) with 500 mg/kg MPI 8. No abnormalities were seen in the heart, lungs, liver and kidneys of mice administered with 500 mg/kg MPI 8 compared to the saline control.

Comment #6: What small molecule phosphate containing compounds are present in vivo?

Response: In light of this comment, we have measured the interaction of MPI 8 with one phosphate containing biomolecule, adenosine diphosphate (ADP). ADP is known to activate platelets in plasma. We have measured the activity of ADP in presence of MPI 8. Our hypothesis was that if ADP is interacting with MPI 8, its activity would decrease. Our results show that MPI 8 does not change the activity of ADP at different concentrations. This data supports our conclusions that MPI 8 is selective against polyP. Please see page 12 (discussion section) and supplementary information figure S11.

Figure S11: Platelet activation when ADP and MPI-8 are added together in platelet rich plasma. MPI-8 was added at different concentrations to ADP and were incubated with platelet rich plasma (PRP). Platelet activation is measured using flow cytometry by measuring the expression of activation marker CD-62P using anti-CD62P-PE antibody. No platelet activation was observed when MPI-8 alone was incubated with PRP at any concentration.

Reviewer #3 (Remarks to the Author):

Comment: This is a concise, well compiled, and written manuscript comprising the in-vitro and in-vivo characterization of macromolecular polyanion inhibitors. The quality of the data is appropriate and technically sound. Presentation and interpretation of the data as well as controls are in large parts of the manuscript sufficient. The presented data reveal sufficiently strong evidence for the author's claims. The report is of interest for the community and can be published – according to the opinion of this reviewer – with minor changes as depicted in the following comments.

Response: We thank the reviewer for carefully reading the manuscript and highly positive comments on our manuscript.

Comment #1: According to the ISO norm - e.g. 10993-4, Biological evaluation of medical devices - Part 4: Selection of tests for interactions with blood - hemocompatibility testing includes analyses of blood platelet and leucocyte as well as complement activation. None of these recommended tests were conducted in this study. Thus, I recommend specifying the terms used to describe the conducted analyses more precisely. The same applies to "Biocompatibility".

The authors state "superior hemocompatibility". In my very personal opinion, superlatives should be avoided in a scientific manuscript to ensure an appropriate scientific discussion.

Response: We have revised our manuscript and removed the use of superlatives. In addition to the studies described, we plan to perform leukocyte and complement activation studies on the lead molecule in a future publication, in order to have a complete description of hemocompatibility of the molecule.

Comment #2: Please describe which mechanisms might be responsible for the reduced bleeding time (MPI 8 – 100, MPI 8, 300) in the mouse model. The differences are not significant, however, a tendency is clearly indicated.

Response: Considering this comment, we have performed additional bleeding measurements using saphenous vein hemostasis model in mice with MPI 8. Our data given in Figure 6 supports that MPI 8 does not increase or decrease bleeding time suggest that this molecule is not changing hemostasis.

One of the mechanisms by which conventional polycations such as protamine, increase bleeding risk by decreasing the thrombin generation by inhibiting Factor V activation. MPI 8 did not induce bleeding even at the highest concentrations tested possibly due to the fact it does change function of coagulation proteins. The low charge density of MPI 8 at physiological pH due to the smart protonation behavior of attached CBG II and the placement of CBGs within the engineered mPEG brush on the polymer scaffold that acts to limit nonspecific binding increases its biocompatibility and selectivity.

We have added this information in the results and in discussion part. Please page 12.

Figure 5. MPI 8 does not induce bleeding in mice. A) High doses of the lead MPI candidates do not cause bleeding in mice. A1) Schematic representation of mouse tail bleeding model with C57/BL6 mice. A2) Recorded bleeding time. Mice were injected in a blinded study with up to 300 mg/kg of lead MPI candidates and demonstrated no increase in bleeding side effect, in contrast to mice administered 200 U/kg of unfractionated heparin. n = 8 mice per group. A3) Hemoglobin lost by mice. B) MPI 8 did not decrease platelet recruitment and fibrin formation in hemostatic clot formation assessed via saphenous vein hemostasis model. Quantitative analysis of the dynamics of platelet accumulation (B1) and fibrin formation (B2) in response to vascular injury in saphenous vein. B3) Representative images of platelet (green) and fibrin (red) hemostatic clot formation in response to a repetitive vascular injury of the saphenous vein. Mice were injected with 200 mg/kg of the lead MPI 8 candidate.

Comment #3: Please discuss in the manuscript the increase in the LDH values as well (Fig 5 C3). Despite differences are tendential only, the data show an outlier in the saline control, which might be the reason why differences between the samples are not significant.

Response: Please see the new figure 7. The measurement of LDH can be sometimes tricky due to hemolysis in the blood sample. The variation might therefore be due to the slight contamination of the sample. The amount of LDH measured is within the normal levels shown by mice of this age. Our histology data also supports the notion there is no tissue damage. Further, we are hesitant to add an additional explanation for this data as the differences were not statistically significant. We plan to perform additional tolerance experiments to support this data in future.

We have re-analyzed the data after removing the outlier, which show no significant difference between the conditions.

We have added this information in the results.

Comment #4: Abbreviations such as ALT and LDH are not explained in the main text. Since the Material and Methods section is provide as a Supplement, I recommend including the explanations in the main text as well.

Response: We have added this information in the main text.

Comment #5: Please correct the error message in the 1.1.1 Materials and Reagents: The protocols for synthesis of UHRA 8, UHRA 10 and MPIs studied are provided in 1.1.2 and the physical characteristics are shown in Error! Reference source not found. BD Vacutainer® Citrate Tubes containing 3.2 % buffered sodium citrate solution were purchased from Becton, Dickinson and Company (New Jersey, USA).

Response: We apologize for the error. We have corrected this in the revised manuscript.

Comment #6: Section 1.1.1.3 Blood collection and plasma preparation The authors state “healthy volunteer donors”. How was the health status of the human donors determined? Which were the inclusion criteria for the donors? Which tests were conducted to exclude inflammatory processes and appropriate hemostatic/cell function? Were the donor data compared with reference values for apparently healthy humans? The number of donors included in the study (not pooled plasma) should be stated in the materials and methods section.

Some of these data are already in the Reporting Summary. I recommend stating this in the manuscript as well.

Characterization of the donors should include the collection of demographic data, data about the recent intake of medication, special nutrition and extensive physical activity. Participants should be excluded from the study if any of these is reported to affect the hemostatic system such as platelet function inhibitors, diabetic and allergic disorders and respective medication as well as excessive consumption of certain food or extensive sports. For thrombogenicity and hemocompatibility studies, characterization of the donated blood should comprise e.g., whole blood haemogram data, analysis of markers of acute inflammation, e.g., C-reactive protein levels as well as test for platelet function for achieving reproducible test results.

Response: We have added any available information as best as possible in the methods section of the revised manuscript. Our experiments involved only in vitro testing. The usual approach for collecting plasma (or whole blood) for in vitro experiments is to utilize volunteers who report no significant serious illnesses, and no recent use of medications that effect platelet or coagulation activities. Plasma is often pooled to overcome any individual abnormalities that might skew results. Minimal demographic data is sometimes collected to provide to the Institutional Review Board and/or overseeing funding agencies. Note that as this was not a clinical study, the extensive data that the reviewer has requested was not necessary and consequently not collected.

REVIEWER COMMENTS

Reviewer #1 (Remarks to the Author):

The authors have followed most of my recommendation and have improved the new version of the manuscript. Specifically, the combination of physicochemical studies with the biological assays have rendered an uncommon study that contains very interesting results, which has the potential to attract the attention of the wide readership of Nature Communications. I am now supporting acceptance of the manuscript in the current form.

Reviewer #2 (Remarks to the Author):

The authors have addressed most of my comments. However, I have several comments about the IVC stenosis data.

1/ What protocol was used? The 2 references provide are a review and a paper that describes an IVC stasis model and an IVC electrolytic model. A detailed protocol should be provide because there are many variations of this model. Were side and/or back branches ligated or cauterized? Was the operator blinded to the groups?

2/ It is stated that IVC ligation was performed approximately 18-24 hours after administration of MPI 8. Why was there such a wide range of times used?

3/ There is a very small range of thrombus weights in the control group for this model.

4/ Twenty mice were used for the experiment (10 per group). Thrombi were observed in 8/10 per group. However, the figure legend states that 2 outliers were removed. This is not consistent with the presence of 8 data points per group. It would be helpful to know the values of these 2 outliers and which group(s) there belonged to.

Response to reviewer's comments:

REVIEWER COMMENTS

Reviewer #1 (Remarks to the Author):

The authors have followed most of my recommendation and have improved the new version of the manuscript. Specifically, the combination of physicochemical studies with the biological assays have rendered an uncommon study that contains very interesting results, which has the potential to attract the attention of the wide readership of Nature Communications. I am now supporting acceptance of the manuscript in the current form.

Response: Thank you very much for re-reading our manuscript and the support.

Reviewer #2 (Remarks to the Author):

The authors have addressed most of my comments. However, I have several comments about the IVC stenosis data.

Response: Thank you very much for the support for this manuscript. We very much appreciate your comments, and in the following paragraphs we address your individual comments.

Comment #1/ What protocol was used? The 2 references provide are a review and a paper that describes an IVC stasis model and an IVC electrolytic model. A detailed protocol should be provide because there are many variations of this model. Were side and/or back branches ligated or cauterized? Was the operator blinded to the groups?

Response: In light of this comment. We have added additional information in the experimental section and added an additional reference.

Kimball AS, Obi AT, Luke CE, Dowling AR, Cai Q, Jankowski H, Schaller M, Jaffer FA, Kunkel S, Gallagher K, Henke PK. Ly6Clo monocyte/macrophages are essential for thrombus resolution in a murine model of venous thrombosis. *Thrombosis Haemostasis* **2020**;120:289-99. PMID 31887775

This reference has been added as Ref. 47 in the main text.

'Twenty male C57BL/6 mice were used to investigate the effect of MPI 8 on venous thrombogenesis. Alzet 1003D micro osmotic pumps (Cupertino, CA) were placed in a subcutaneous location to deliver MPI 8 (n=10) or control vehicle (n=10) at a rate of 100 ug/hr approximately 18-24 hours prior to venous thrombosis (VT) induction. The St. Thomas Stenosis Model to induce VT was performed using a 30g needle spacer and ligation of side branches, but not back branches, as described (*Reference: PMID 30786739 and PMID 31887775*). The dosage times vary because the installation time of the Alzet pump is much less than the stenosis surgery time. Harvest of the thrombosed IVC segment for analysis was at 48 hours.'

The surgeon was not blinded for the study.

This information has been added in the methods section (supplementary information)

Comment #2/ It is stated that IVC ligation was performed approximately 18-24 hours after administration of MPI 8. Why was there such a wide range of times used?

Response: See the response to the comment # 1 above. Since we have used micro osmotic pumps to administer the MPI-8. To achieve the steady state of MPI-8 and to avoid the complication of inserting the pump before venous thrombosis (VT) induction. This has been clarified in the experimental details.

Comment #3/ There is a very small range of thrombus weights in the control group for this model.

Response: Thanks for this comment. Based on our experience, the weight of the thrombus is not unusual. The procedure is performed by a highly experienced surgeon with experience in this model. We are highly confident of the data provided.

Comment #4/ Twenty mice were used for the experiment (10 per group). Thrombi were observed in 8/10 per group. However, the figure legend states that 2 outliers were removed. This is not consistent with the presence of 8 data points per group. It would be helpful to know the values of these 2 outliers and which group(s) they belonged to.

Response: The outliers removed were not data outliers, they were model outliers where no clot was formed in the IVC. The stenosis model has variable clot generation due to flow in the IVC. The observation is quite natural. This has been explained in the figure caption.

Figure caption was modified as

Fig 5: C) MPI 8 treatment inhibits thrombus formation in inferior vena cava thrombosis model. C1) a cartoon representation of the mouse model. C2) Thrombus weights of untreated (vehicle control) and treated (MPI 8) mice. After removal of 2 outliers per group, **due to a lack of clot formation**, data were statistically significantly different at $p = 0.0003$ in an unpaired T-test with Welch's correction.

REVIEWERS' COMMENTS

Reviewer #2 (Remarks to the Author):

The authors have addressed my concerns.